# Binding of NUFIP2 to Roquin promotes recognition and regulation of *ICOS* mRNA

Nina Rehage[1,2], Elena Davydova[3], Christine Conrad[1], Gesine Behrens[1], Andreas Maiser[4], Jenny E. Stehklein[2], Sven Brenner[2], Juliane Klein[1], Aicha Jeridi[2], Anne Hoffmann[5,14], Eunhae Lee[6,7], Umberto Dianzani[8], Rob Willemsen[9], Regina Feederle[10], Kristin Reiche[11], Jörg Hackermüller[5], Heinrich Leonhardt[4], Sonia Sharma[6,7], Dierk Niessing [3,12,13] & Vigo Heissmeyer[1,2]

The ubiquitously expressed RNA-binding proteins Roquin-1 and Roquin-2 are essential for appropriate immune cell function and postnatal survival of mice. Roquin proteins repress target mRNAs by recognizing secondary structures in their 3′-UTRs and by inducing mRNA decay. However, it is unknown if other cellular proteins contribute to target control. To identify cofactors of Roquin, we used RNA interference to screen ~1500 genes involved in RNA-binding or mRNA degradation, and identified NUFIP2 as a cofactor of Roquin-induced mRNA decay. NUFIP2 binds directly and with high affinity to Roquin, which stabilizes NUFIP2 in cells. Post-transcriptional repression of human ICOS by endogenous Roquin proteins requires two neighboring non-canonical stem-loops in the *ICOS* 3′-UTR. This unconventional *cis*-element as well as another tandem loop known to confer Roquin-mediated regulation of the *Ox40* 3′-UTR, are bound cooperatively by Roquin and NUFIP2. NUFIP2 therefore emerges as a cofactor that contributes to mRNA target recognition by Roquin.

[1] Institute for Immunology at the Biomedical Center, Ludwig-Maximilians-Universität München, Grosshaderner Strasse 9, 82152 Planegg-Martinsried, Germany. [2] Research Unit Molecular Immune Regulation, Helmholtz Zentrum München, Marchioninistrasse 25, 81377 München, Germany. [3] Group Intracellular Transport and RNA Biology, Institute of Structural Biology, Helmholtz Zentrum München, Ingolstädter Landstrasse 1, 85764 Neuherberg, Germany. [4] Center for Integrated Protein Science at the Department of Biology, Ludwig-Maximilians-Universität München, Grosshaderner Strasse 2, 82152 Planegg-Martinsried, Germany. [5] Young Investigators Group Bioinformatics and Transcriptomics, Department Molecular Systems Biology, Helmholtz Centre for Environmental Research – UFZ, Permoserstraße 15, 04318 Leipzig, Germany. [6] Division of Cell Biology, La Jolla Institute for Allergy and Immunology, 9420 Athena Circle, La Jolla, CA 92037, USA. [7] The Functional Genomics Center, La Jolla Institute for Allergy and Immunology, 9420 Athena Circle, La Jolla, CA 92037, USA. [8] Department of Health Sciences, Universita' del Piemonte Orientale, via Solaroli 17, 28100 Novara, Italy. [9] CBG Department of Clinical Genetics, Erasmus MC, Wytemaweg 80, 3015 CN Rotterdam, Netherlands. [10] Monoclonal Antibody Core Facility and Research Group, Institute for Diabetes and Obesity, Helmholtz Zentrum München, Marchioninistrasse 25, 81377 München, Germany. [11] Bioinformatic Unit, Department of Diagnostics, Fraunhofer Institute for Cell Therapy and Immunology- IZI, 04103 Leipzig, Germany. [12] Department of Cell Biology at the Biomedical Center, Ludwig-Maximilians-Universität München, Grosshaderner Strasse 9, 82152 Planegg-Martinsried, Germany. [13] Institute of Pharmaceutical Biotechnology, Ulm University, James Franck Ring N27, 89081 Ulm, Germany. [14] Present address: Bioinformatics Group, Department of Computer Science; and Interdisciplinary Center of Bioinformatics, Leipzig University, Härtelstraße 16-18, 04107 Leipzig, Germany. Nina Rehage, Elena Davydova and Christine Conrad contributed equally to this work. Correspondence and requests for materials should be addressed to S.S. (email: soniasharma@lji.org) or to D.N. (email: dierk.niessing@uni-ulm.de) or to V.H. (email: vigo.heissmeyer@med.uni-muenchen.de)

The RNA-binding proteins (RBPs) Roquin-1 and Roquin-2 recognize specific stem-loop structures in the 3′-untranslated regions (3′-UTR) of target mRNAs through their ROQ domain[1–4]. Binding typically results in post-transcriptional repression of target-mRNA expression through 5′ mRNA degradation[5]. However, the importance and cooperation of other RBPs in this function is still elusive. Drosophila, mouse and human ROQUIN-1 and ROQUIN-2 were described to interact with the CCR4-CAF1-NOT de-adenylation complex[5–7]. Deadenylation is then coupled to mRNA decapping, followed by 5′ to 3′-directed mRNA decay[5]. It is likely that Roquin induces post-transcriptional repression as part of higher-order messenger ribonucleoprotein particles (mRNPs) that can be regulated in cell-type specific and dynamic ways and differ among the cellular target mRNAs. Especially on long and complex 3′-UTRs, Roquin may interact, synergize or interfere with other post-transcriptional regulators that work in a redundant, cooperative or antagonistic way. Indeed, the 3′ terminal 260 nucleotides (nts) of the *TNF* 3′-UTR were sufficient to mediate repression by Roquin-1 and the endonuclease Regnase-1 in a cooperative manner[8], while other target mRNAs may be repressed by each *trans*-acting factor individually[9]. Except for Regnase-1, crucial cofactors and collaborating RBPs of Roquin are still unknown.

The two Roquin paralogs are expressed ubiquitously and are also required in cells outside of the adaptive immune system, since mice with systemic deletion of Roquin-1 or Roquin-2 die postnatally[10,11]. Both proteins serve redundant functions in T cells[11] by regulating T-cell activation and T-helper cell differentiation[8,11–13]. The critical importance of Roquin-mediated gene regulation in T cells is underscored by its cleavage by MALT1 downstream of T-cell receptor signaling, which affects multiple target mRNAs with T-cell-specific expression and function[8,12,14]. Accordingly, the inducible T-cell co-stimulatory receptor *ICOS*, whose ~2000 nucleotide-long 3′-UTR harbors *cis*-elements for several *trans*-acting factors[8,15–17], was the first identified mRNA target of Roquin-1[13]. In *sanroque* mice, a point mutation in the ROQ domain of Roquin-1 impairs Roquin function and causes derepression of ICOS already in naive T cells[13]. It has been proposed that inappropriate ICOS expression can explain the development of the severe autoimmunity of *sanroque* mice[18] although additional deletion of ICOS did not suffice to rescue autoimmunity[19]. Nevertheless, ICOS is important for the expansion and survival of regulatory T cells (Tregs) and effector memory T cells[20]. ICOS signals are also required for the differentiation of follicular helper T cells (Tfh) and germinal center B cells[21,22]. ICOS stimulation induces PI3K activity and Foxo-1 inactivation[23] and was shown to recruit activated CD4$^+$ T cells into the follicle[24] and to be required for the maintenance of a germinal center response[25]. Finally, patients with loss-of-function mutations in ICOS are immunodeficient[26]. The principles of post-transcriptional regulation of *ICOS* are therefore of considerable interest and the underlying molecular mechanisms may similarly control other, perhaps even unknown mRNA targets of Roquin proteins.

In this study, we identify NUFIP2 as an important cofactor of Roquin-mediated post-transcriptional gene regulation of *ICOS*. Investigating its functional contribution, we demonstrate that NUFIP2 engages in a direct, high-affinity interaction with Roquin-1 that enables cooperative binding of this complex to tandem stem-loops in the *ICOS* and *Ox40* 3′-UTRs. Our data indicate cofactor-dependent target specificity in Roquin-mediated post-transcriptional gene regulation.

## Results

**Targeted siRNA screening to identify cofactors of Roquin**. To search for potential cofactors of Roquin-mediated post-transcriptional gene regulation, we performed a targeted siRNA screen. In a HeLa reporter cell line stably co-expressing ICOS and an inducible Roquin-1-P2A-mCherry open reading frame (Fig. 1a, b), we observed strong downregulation of ICOS protein levels after doxycycline-induced Roquin-1 and mCherry expression (Fig. 1b). siRNA-mediated depletion of Roquin resulted in derepression of ICOS (Fig. 1c, d). The assay was both robust and reproducible, as indicated by a Z′ factor of 0.7 (Fig. 1d), which reflects the suitability of a given assay for high-throughput screening[27]. In the screen, each gene was targeted by a pool of four individual siRNAs, and multiplexed flow cytometry was employed during data acquisition (Supplementary Fig. 1a, b). In total, we screened 1495 siRNAs specifically targeting genes encoding proteins associated with four ontology terms: "RNA-binding proteins"[28], "P body/stress-granule-related proteins"[29], "deadenylation-dependent mRNA decay", and "decapping-dependent mRNA decay"[30–32] (Supplementary Data 1, Fig. 1e). EDC4 and CNOT1, two effectors of mRNA deadenylation and decapping that interact with Roquin[6,15], scored positively and thereby validated our screen. Yet, REGNASE-1, an endo-ribonuclease that cooperates with Roquin-1 in the repression of a reporter containing the most 3′ located 260 nts of the *TNF* 3′-UTR (termed CDE$_{260}$)[8], was not identified in our screen. Investigating why REGNASE-1 (encoded by the *ZC3H12A* gene) was not a hit in this screen, we found that the regulation of the *ICOS* 3′-UTR by Roquin-1 did not depend on Regnase-1, in contrast to the CDE$_{260}$ *cis*-element of the *TNF* 3′-UTR (Supplementary Fig. 1c, d)[8]. Specifically, Roquin-1 overexpression downregulated the ICOS reporter to a similar extent in Regnase-1-deficient (*Zc3h12a$^{-/-}$*) (Supplementary Fig. 1f) and Roquin-deficient (*Rc3h1/2$^{-/-}$*) mouse embryonic fibroblast (MEF) cells (Supplementary Fig. 1e). Moreover, the *ICOS* 3′-UTR was similarly regulated by overexpression of Regnase-1 in Roquin-deficient and Regnase-1-deficient cells (Supplementary Fig. 1e, f). Together, these results show that the screen identified known genes involved in Roquin-mediated ICOS regulation as well as new candidates.

**Validation of NUFIP2 as a cofactor of Roquin**. We validated top scoring candidates in the siRNA screen by "deconvoluting" the siRNA pools and testing each individual siRNA. Candidates such as *STAUFEN* (*STAU1*) did not pass this validation step (Supplementary Data 1), as only one siRNA reversed ICOS repression (Supplementary Fig. 2a, b), despite the fact that all four siRNAs efficiently depleted *STAU1* mRNA (Supplementary Fig. 2c). In contrast, these analyses confirmed CNOT1 as a positive control and validated NUFIP2 as a cofactor of Roquin-1-mediated ICOS regulation. For these two targets, multiple siRNAs from the original pool decreased *CNOT1* or *NUFIP2* target mRNA without diminishing Roquin-1-P2A-mCherry expression and caused derepression of *ICOS* (Fig. 2a–c). Our *CNOT1* results confirm the recently demonstrated impairment of *Dm* Roquin to induce reporter mRNA degradation in *Drosophila* cells with CNOT1 depletion[5]. Strikingly, NUFIP2 was among the strongest hits in this screen, and its knockdown derepressed ICOS more effectively than that of *CNOT1* (Fig. 3a). Both siRNA pools efficiently decreased target gene expression on mRNA and protein levels (Fig. 3b, c). We further validated NUFIP2 as a cofactor of Roquin through rescuing the effect of the siRNA by over-expressing an *siRNA#2*-resistant NUFIP2 cDNA (Supplementary Fig. 2d). GFP-NUFIP2- or GFP-overexpressing HeLa reporter cells (Fig. 3d) were transfected with *siRNA#2*, which only targets the endogenous protein, or *siRNA#4*, which targets both endogenous and ectopically expressed NUFIP2 (Supplementary Fig. 2d). ICOS surface expression upon doxycycline-induced

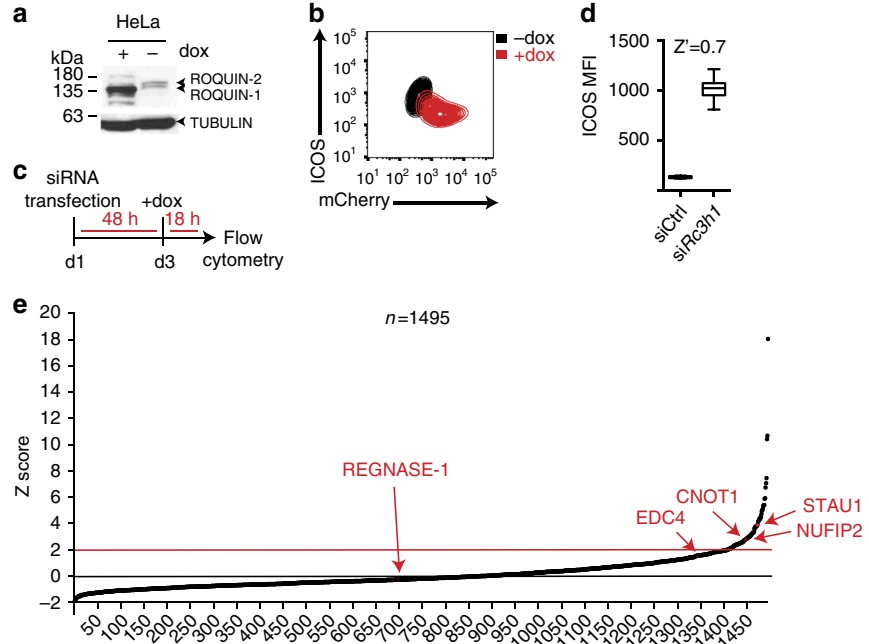

**Fig. 1** A targeted siRNA screen to identify cofactors of Roquin-mediated post-transcriptional gene regulation. **a** Immunoblot analysis of Roquin-1, Roquin-2, and α-Tubulin expression or **b** flow cytometry of ICOS and mCherry expression in HeLa reporter cells containing cassettes for stable ICOS and doxycycline-inducible Roquin-1-P2A-mCherry overexpression. Cells were either treated with doxycycline (dox) for 18 h or left untreated. **c** Schematic representation of the screen workflow. **d** Distribution of ICOS mean fluorescence intensity (MFI) in HeLa reporter cells after transfection with Roquin-1-targeting siRNA pools (*siRc3h1*) or non-targeting control siRNAs (*siCtrl*) in a 96-well plate. The $Z'$ factor was calculated from mean and SDs of positive (*siRc3h1*) and negative (*siCtrl*) control data. **e** Normalized screen data of the customized siRNA library for Roquin-1 cofactors. ICOS MFI of each sample was normalized into a $Z$ score based on plate mean and SD. Ranked $Z$ scores are shown for each siRNA pool. Each data point with an average $Z$ score >2 was considered a hit

Roquin-1 overexpression was analyzed in gates that contained low (GFP^lo), medium (GFP^med), or high (GFP^hi) GFP expression (Fig. 3e, f). Importantly, in siRNA#2-transfected cells, ICOS was significantly downregulated by increasing expression of siRNA#2-resistant GFP-NUFIP2 in GFP^med and GFP^hi cells (Fig. 3f, upper graph). Cells with comparable GFP-NUFIP2 expression were not observed when GFP-NUFIP2 expression was combined with transfections of *siRNA#4* that effectively reduced the ectopic NUFIP2 expression (Fig. 3e). In the remaining GFP-expressing cells within the GFP^med gate the ICOS repression was also not rescued (Fig. 3f, lower graph). Of note, the transduction of cells with GFP did not change ICOS expression (Fig. 3f upper and lower graph). Together these experiments identified and validated NUFIP2 as a cofactor of Roquin-mediated ICOS repression.

We next asked how ICOS was regulated by NUFIP2 and whether this regulation required Roquin proteins. To do so we combined the siRNA against NUFIP2 with those against RC3H1 and RC3H2 to target both ROQUIN-1- and ROQUIN-2-encoding mRNAs in HeLa cells that stably expressed ICOS mRNA (Fig. 3g, h). The combinations of all three siRNAs equally reduced NUFIP2 and ROQUIN-encoding mRNAs compared to transfections that targeted the factors individually (Fig. 3g). Treating these cells with actinomycin D and measuring ICOS mRNA abundance by quantitative RT-PCR over time revealed a half-life of ICOS mRNA of 158 min in control siRNA-treated cells, which increased to 232 min when NUFIP2 was knocked down. Importantly, the knockdown of ROQUIN- or ROQUIN- and NUFIP2-encoding mRNAs similarly stabilized the ICOS mRNA to a half-life of 345 or 360 min, respectively (Fig. 3h). This indicates that NUFIP2 cooperates with ROQUIN to induce ICOS mRNA decay, as the effectiveness of NUFIP2 to induce

degradation of *ICOS* mRNA depended on the presence of ROQUIN proteins (Fig. 3h).

**Molecular determinants of NUFIP2 function.** For the detection of NUFIP2, we established two different monoclonal antibodies, 23G8 and 14G9, for western blotting and immunoprecipitation (Supplementary Fig. 2e, f) or intracellular staining and flow cytometry, respectively (Supplementary Fig. 2g, h).

Roquin function is associated with its localization in RNA granule-like structures. In the absence of cell stress, the protein is enriched in P bodies[15]. In response to arsenite-induced oxidative stress, Roquin relocalizes to stress granules[15,33,34]. As a biochemical assay that mimics aggregation of proteins in granule-like structures, the chemical biotinylated isoxazole (b-isox) precipitates constituents of stress granules like FMRP, FXR1P and FXR2P from cell lysates[35]. Since NUFIP2 has been described to interact with FMRP in cells[36], we investigated aggregation of Roquin, Nufip2, and Fmrp by incubating MEF cell lysates with increasing concentrations of b-isox. Treatment and subsequent centrifugation localized the Fmrp protein to the pellet, depleting it from the supernatant at all b-isox concentrations (Fig. 4a). Nufip2 precipitated at 1 μM b-isox, but unlike Fmrp was not depleted from the supernatant. The Roquin proteins also partially precipitated, but only in response to the high concentrations of b-isox (Fig. 4a). Consistent with the presence of low complexity regions (LCR) in the carboxy-terminus of Roquin, the amino-terminal fragment of Roquin, which results from MALT1 paracaspase-mediated cleavage[12,14], did not precipitate as well (Fig. 4a). These experiments suggest that in fibroblast extracts only a subset of the Nufip2 protein is associated

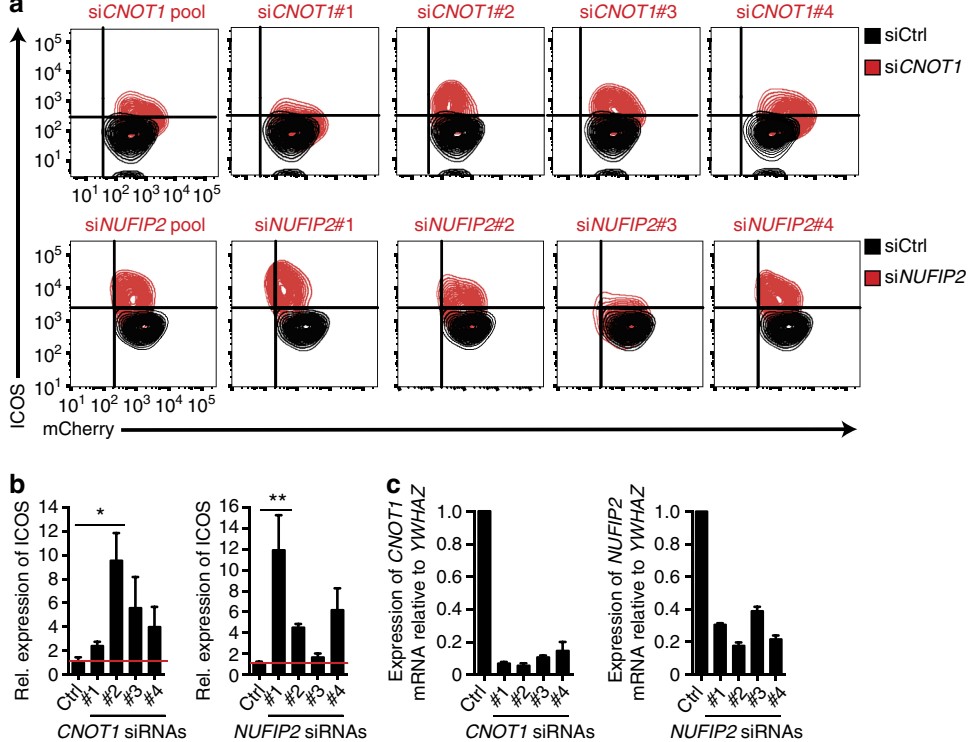

**Fig. 2** Deconvolution of siRNA pools to validate genuine hits. **a** Flow cytometry analysis of ICOS and mCherry expression in HeLa reporter cells treated with individual or pooled CNOT1 or NUFIP2 siRNAs (red) or non-targeting control siRNAs (siCtrl) (black). Prior to analysis, cells were treated with doxycycline for 18 h to induce Roquin-1 overexpression. **b** Quantified ICOS MFI of cells shown in **a**. Expression levels were normalized to siCtrl-treated cells. **c** qPCR analysis of *CNOT1* or *NUFIP2* mRNA expression in cells from **a**. Expression was calculated relative to the reference gene *YWHAZ* and normalized to siCtrl-treated cells. Error bars in **b** and **c** represent mean and SD of three independent experiments. In **a** representatives of three independent experiments are shown. Statistical significance in **b** was calculated with one-way ANOVA Kruskal–Wallis test followed by Dunn's multiple comparisons test (*$p < 0.05$, **$p < 0.01$)

with Fmrp, and that localization of Roquin into RNA granule-like structures is likely independent of Fmrp or Nufip2.

We next investigated the cellular localization of NUFIP2 to RNA granule-like structures by super-resolution microscopy in MEF cell lines and flow cytometry imaging in primary CD4[+] T cells (Fig. 4b, Supplementary Figs. 3, 4, 5a–e). Although NUFIP2 was named nuclear FMRP interacting protein 2 due to its nucleo-cytoplasmic relocalization during different phases of the cell cycle[36], we found retrovirally expressed GFP-NUFIP2 to be predominantly localized in the cytoplasm of T cells (Fig. 4b). Cytoplasmic localization of NUFIP2 was observed throughout all phases of the cell cycle (Fig. 4b and Supplementary Fig. 3a–c), and was unchanged in the absence of Roquin (Supplementary Fig. 3d–f). As described previously[37], both endogenous and overexpressed NUFIP2 aggregated into stress granules in response to arsenite-induced oxidative stress (Supplementary Figs. 4a, b, 5a–c). However, we observed a partial enrichment of NUFIP2 in P bodies (Supplementary Figs. 4a, 5a, c), where it colocalized with Roquin-1 (Supplementary Fig. 4a). Neither absence of NUFIP2 affected Roquin-1 localization to P bodies or stress granules (Supplementary Fig. 5d, e), nor did the absence of Roquin interfere with aggregation of NUFIP2 in stress granules (Supplementary Fig. 4b). However, the specific enrichment of NUFIP2 in P bodies (Supplementary Figs. 4a, 5c) was not observed anymore in the absence of Roquin expression (Supplementary Fig. 4b).

We investigated the expression profile of Nufip2 in mouse tissues and upon T-cell stimulation. Nufip2 was expressed highest in the brain, but also enriched in spleen, lymph node, thymus, and lung compared to lower expression levels in heart, muscle,

kidney, and liver (Fig. 4c). Notably, highest expression of Nufip2 protein correlated with high Roquin levels[11], and T-cell stimulation significantly increased protein expression of Nufip2 and Roquin-1 and Roquin-2 (Fig. 4d). This effect was not increased further by addition of agonistic antibodies against the costimulatory receptors CD28 or ICOS (Fig. 4d). In contrast, stimulation with the cytokines IL-2, IL-6 and IL-10, the latter of which has been shown to induce Roquin-1 mRNA expression[38], was not able to elevate either Nufip2 or Roquin-1 and Roquin-2 protein levels under conditions that effectively induced phosphorylation of Stat3 or Stat5 (Supplementary Fig. 5f). Analyzing Nufip2 protein amounts in peripheral CD4[+] T cells, we found that deletion of the Roquin encoding alleles also caused a reduction in the Nufip2 protein (Fig. 4e), which could not be accounted for by a comparably small decrease of *Nufip2* mRNA levels (Supplementary Fig. 5g). Accordingly, Nufip2 expression was restored in Roquin-deficient MEF cells upon retroviral reconstitution with Roquin-1 (Fig. 4f), while mRNA levels stayed constant (Fig. 4g). Interestingly, rescue of Nufip2 expression occurred when cells were reconstituted with wild-type Roquin, but also upon expression of RNA-binding (Roquin[K220A K239A R260A]) or post-transcriptional inactive (aa 1–509) Roquin-1 mutants, or upon expression of the *sanroque* mutant of Roquin-1, Roquin-1[M199R] (Fig. 4f). These results suggest that Roquin stabilizes Nufip2 on the post-translational level.

**Direct physical interaction between Roquin and NUFIP2.** Since NUFIP2 was destabilized in the absence of Roquin, we tested whether both proteins interact. Overexpressed NUFIP2 co-

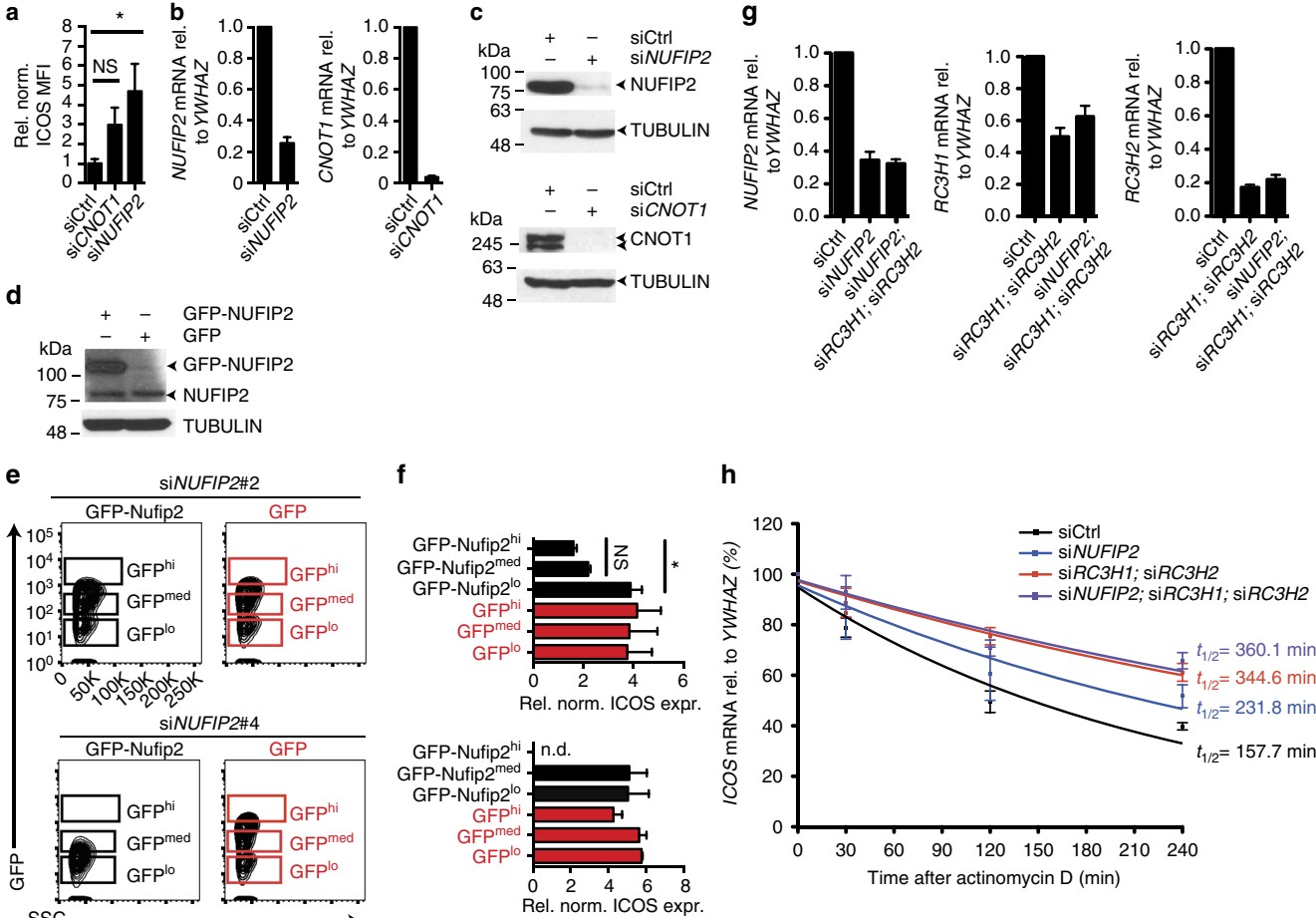

**Fig. 3** ICOS repression by NUFIP2 depends on Roquin expression. **a** MFI of ICOS expression or **b** qPCR analysis of *NUFIP2* and *CNOT1* relative to *YWHAZ* mRNA expression after siRNA knockdown of *NUFIP2* or *CNOT1* in HeLa reporter cells using siGENOME siRNA pools. **c** Immunoblot analysis of NUFIP2 and CNOT1 expression in HeLa reporter cells after treatment with the siGENOME *NUFIP2* or *CNOT1* siRNA pool, respectively. **d** Immunoblot analysis of NUFIP2 expression in reporter cells that were transduced with GFP- or GFP-NUFIP2 by retroviral infection. **e** Flow cytometry analysis of GFP expression in HeLa reporter cells that were transduced with different amounts of GFP- or siRNA-resistant GFP-NUFIP2 by retroviral infection. **f** GFP- or siRNA-resistant GFP-NUFIP2-expressing reporter cells from **e** were transfected with *NUFIP2*-targeting or control siRNAs and Roquin expression was induced with doxycycline. ICOS expression was measured by flow cytometry and normalized to siCtrl-treated cells. ICOS expression in cells with high (hi) or intermediate (med) GFP-expression is compared with that of low (lo) GFP-expressing cells. **g** qPCR analysis of *NUFIP2*, *RC3H1*, and *RC3H2* mRNA expression in HeLa reporter cells after transfection with the indicated siRNAs. Expression relative to the reference gene *YWHAZ* was normalized to the non-targeting control. **h** *ICOS* mRNA decay curves of HeLa reporter cells after treatment with the siGENOME *NUFIP2*-targeting siRNA pool, a combination of siRNAs against *RC3H1* and *RC3H2* (Invitrogen) or a combination of all three targets or a non-targeting control siRNA. Forty-eight hours after transfection, cells were seeded for treatment with 5 μg/mL actinomycin D the next day. Cells were treated for 30–240 min and *ICOS* expression was determined by qPCR analysis and normalized to *YWHAZ*. In **c** and **e** representatives of three independent experiments are shown. Error bars in **a**, **b**, **d**, **f**, **g**, and **h** represent mean and SDs of three (**a**, **b**, **f**, **g**) or four (**d**, **g**, **h**) independent experiments. mRNA half-life in **h** was calculated with Graph Pad Prism from four independent experiments. In **a** and **f** statistical significance was calculated with one-way ANOVA Kruskal–Wallis test followed by Dunn's multiple comparisons test (*$p < 0.05$)

immunoprecipitated with GFP-Roquin-1 from HEK293T cell lysates using anti-GFP antibodies in an RNase-insensitive manner (Fig. 5a). Surprisingly, Roquin-1 interacted with NUFIP2 through the amino-terminus (aa 1–509), harboring the RING finger and ROQ domain, and not through the carboxy-terminal sequences (aa 509–1130) that have been proposed to mediate protein−protein interactions (Fig. 5b and Supplementary Fig. 6a). Underscoring the physiologic importance of this interaction, endogenous Nufip2 was strongly enriched in immunoprecipitates of endogenous Roquin from MEF cell lysates using anti-Roquin-1/2−specific antibodies (Fig. 5c). This interaction still occurred with the *sanroque* mutant of Roquin-1 (Fig. 5c). Notably, immunoprecipitating Nufip2 from lysates of individual Roquin-1 or Roquin-2 knockout cells revealed indistinguishable co-immunoprecipitation with either paralog (Fig. 5d). The NUFIP2

(aa 255–411) internal fragment was essential for the interaction, since neither amino-terminal (aa 1–255) nor carboxy-terminal sequences (aa 411–695) co-immunoprecipitated with Roquin-1 (Fig. 5e, Supplementary Fig. 6a). Finally, we purified recombinant proteins produced in bacteria (Fig. 5f) and demonstrated a direct interaction between mouse Roquin-1 (aa 2–441) and human NUFIP2 (aa 255–411) by surface plasmon resonance (SPR) experiments (Fig. 5g). The calculated equilibrium dissociation constant ($K_D$) of 182 nM indicates a strong physical interaction (Fig. 5h).

Since the same fragment of NUFIP2 that mediated interaction with Roquin has also been shown to engage in a direct interaction with an amino-terminal fragment of Fmrp (aa 1–132), we wondered whether the interaction between NUFIP2 and Roquin and between NUFIP2 and Fmrp were mutually exclusive or

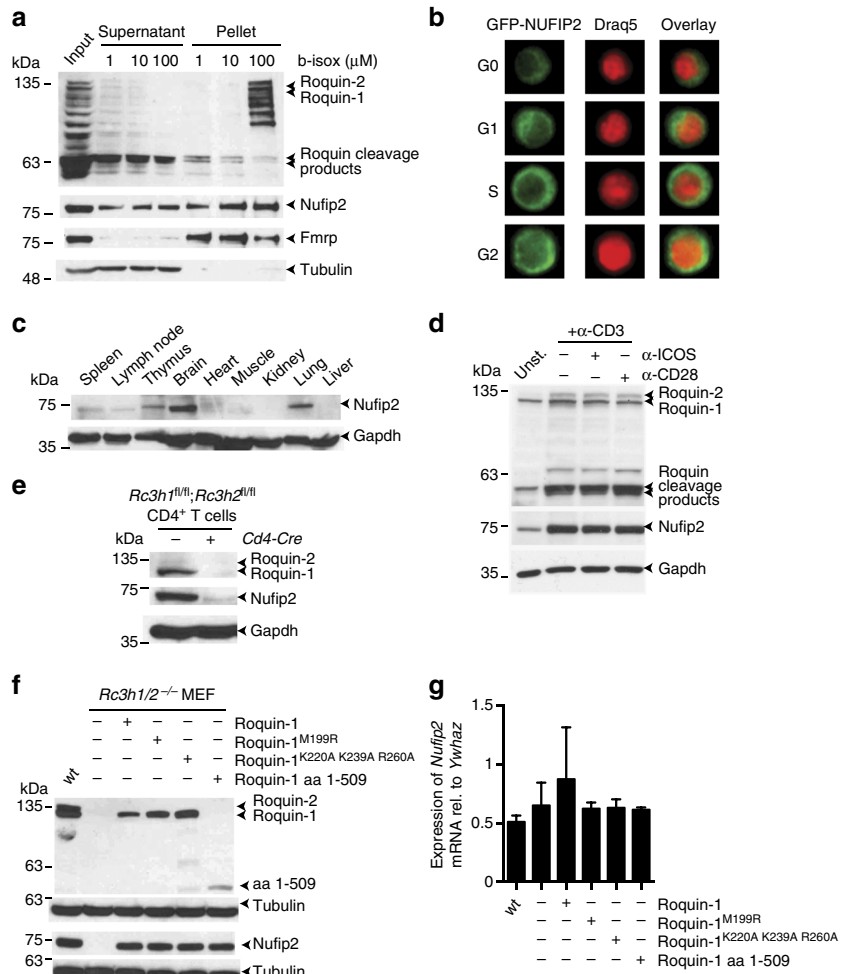

**Fig. 4** Nufip2 protein is stabilized by Roquin. **a** Immunoblot analysis of Roquin, Nufip2, and Fmrp proteins in MEF lysates and supernatants or precipitates of b-isox-treated extracts. **b** Flow cytometry imaging analysis of GFP-Nufip2 localization in activated Th1 cells from *Rc3h1*[fl/fl]; *Rc3h2*[fl/fl]; *Cd4-Cre-ERT2*; *rtTA* mice. Before analysis, retrovirally transduced cells were treated with doxycycline for 24 h to induce GFP-NUFIP2 expression. **c–f** Immunoblot analysis of Nufip2 (**c**) or Nufip2 and Roquin (**d–f**) in **c** different mouse tissues, **d** mouse CD4⁺ T cells left untreated or stimulated for 12 h with (α-CD3) alone or in combination with anti-CD28 (α-CD28) or anti-ICOS (α-ICOS), **e** mouse CD4⁺ T cells with (*Rc3h1/2*[fl/fl]) or without (*Rc3h1/2*[fl/fl]; *Cd4-Cre*) Roquin expression, or **f** in lysates from wild-type and Roquin–deficient MEF cells, reconstituted with wild-type or Roquin-1 mutants as indicated. Gapdh (**c–e**) or α-tubulin (**f**) served as loading controls. **g** qPCR analysis of *Nufip2* mRNA expression in cells from **f**. Expression was calculated relative to the reference gene *Ywhaz* and normalized to untransduced cells. Error bars in **g** represent mean and SD of two independent experiments. In **a**, **b**, **d**, **e** and **f** representatives of two (**d**, **e**, **f**) or three (**a**, **b**) independent experiments are shown

enabled formation of a ternary complex. To answer this question, we purified the fragment of Fmrp (aa1–132) (Fig. 6a) that binds to NUFIP2 (Fig. 6b), and assayed binding of Roquin to NUFIP2 by SPR in the presence of Fmrp protein. Addition of Fmrp inhibited the binding of NUFIP2 (aa 255–411) to immobilized Roquin-1 (aa 2–441) in a concentration-dependent way (Fig. 6b). Consistently, less Roquin was co-immunoprecipitated with Nufip2 from cell lysates when Fmrp was overexpressed in the cells (Fig. 6c). These results demonstrate a mutually exclusive interaction of NUFIP2 with Fmrp or Roquin-1, which likely leads to competition for binding to NUFIP2 by Roquin and Fmrp. However, analyzing prototypical Roquin functions in mice lacking both alleles of the Fmrp-encoding gene *Fmr1* (*Fmr1*⁻/⁻ mice), there was no indication for a gain of Roquin function, since those animals contained comparable frequencies of effector-like (CD62⁻ CD44⁺) or regulatory T cells in secondary lymphoid organs (Fig. 6d, e). Furthermore, ICOS surface expression was not changed in T cells with *Fmr1*⁻/⁻ compared to wild-type genotypes when we analyzed Foxp3⁺ regulatory CD4⁺ T cells (Fig. 6e, f) or

conventional CD4⁺ T cells without (Fig. 6g, 0 day) or with stimulation in a 6-day culture (Fig. 6g).

**Roquin and NUFIP2 cooperatively bind to *cis*-elements.** As previously reported, progressive 3′ shortening of the full-length human *ICOS* 3′-UTR (1–2478) leads to a gradual reduction in the repression of ICOS by overexpressed Roquin-1 (Supplementary Fig. 6b)[15]. This implies the existence of multiple alternative binding sites for overexpressed Roquin-1 in the *ICOS* 3′-UTR. To determine which *cis*-elements in human *ICOS* respond to endogenous Roquin, we evaluated *ICOS* 3′-UTR deletion constructs in a *Rc3h1*[fl/fl]; *Rc3h2*[fl/fl]; *Cre-ERT2* MEF cell line before and after 4′OH-tamoxifen-inducible deletion of Roquin-encoding alleles (Fig. 7a and Supplementary Fig. 6c). Strikingly, derepression occurred when the 3′-UTR of the *ICOS* mRNA was shortened from the 3′ end to 2211 nts or beyond, while constructs with at least 2271 nts were fully repressed by endogenous Roquin. These results suggest that a critical Roquin-regulated *cis*-element is

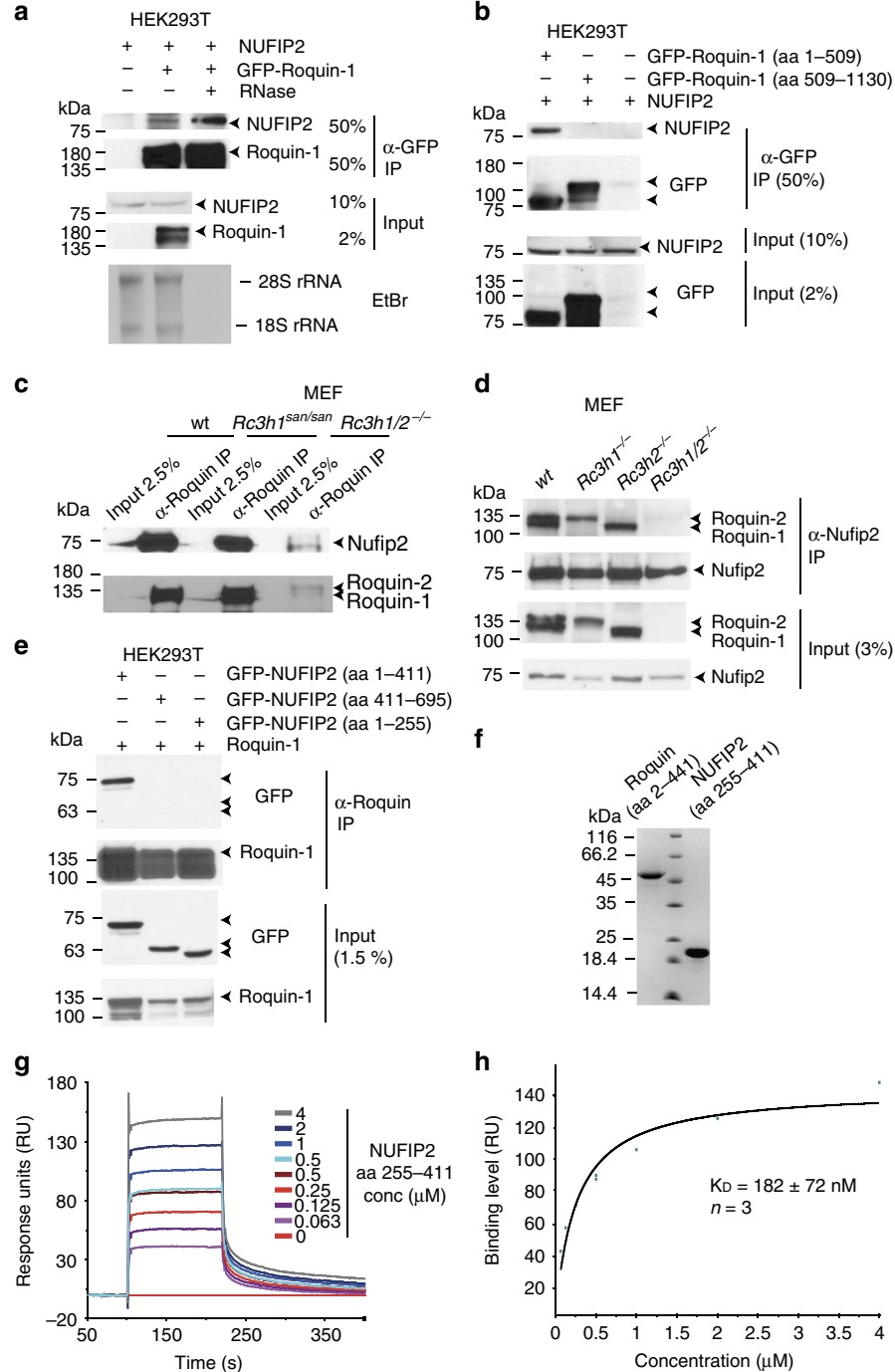

**Fig. 5** Direct physical interaction between Nufip2 and Roquin. Immunoblots were analyzed with antibodies against Nufip2 and Roquin, Roquin and GFP or Nufip2 and GFP. Anti-GFP or anti-Roquin immunoprecipitation from lysates of HEK293T cells transfected with the indicated expression vectors (**a**, **b**, **e**) and anti-Roquin (**c**) or anti-Nufip2 (**d**) immunoprecipitation of endogenous proteins from MEF cells of the indicated genotypes are shown. **a** Extracts were treated during immunoprecipitation with or without RNase as indicated and degradation of rRNA by RNase treatment was confirmed by ethidium bromide (EtBr) staining of RNA extracts from IP supernatants. **f** Representative coomassie-stained PAGE of purified Roquin-1 (aa 2–441) and NUFIP2 (aa 255–411) proteins. **g**, **h** Surface plasmon resonance study of the binding of NUFIP2 (aa 255–411) to immobilized Roquin-1 (aa 2–441). **g** Biacore sensogram recording the binding of NUFIP2 (aa 255–441) injected at two-fold serial dilutions ranging from 0.063 to 4 μM to Roquin-1 (aa 2–441). After the highest concentration, the 0.5 μM sample dilution was re-injected to assess potential accumulation of unspecific background binding. **h** Steady-state affinity analysis of the binding level in response units (RUs) against the Nufip2 (aa 255–411) protein concentration shown in a representative fitting curve. In **a–e** and **g–h** one representative of two (**d**) or three (**a–c**, **e**, **g–h**) independent experiments is shown. The $K_D$ in **h** was calculated from mean and SD of three independent experiments

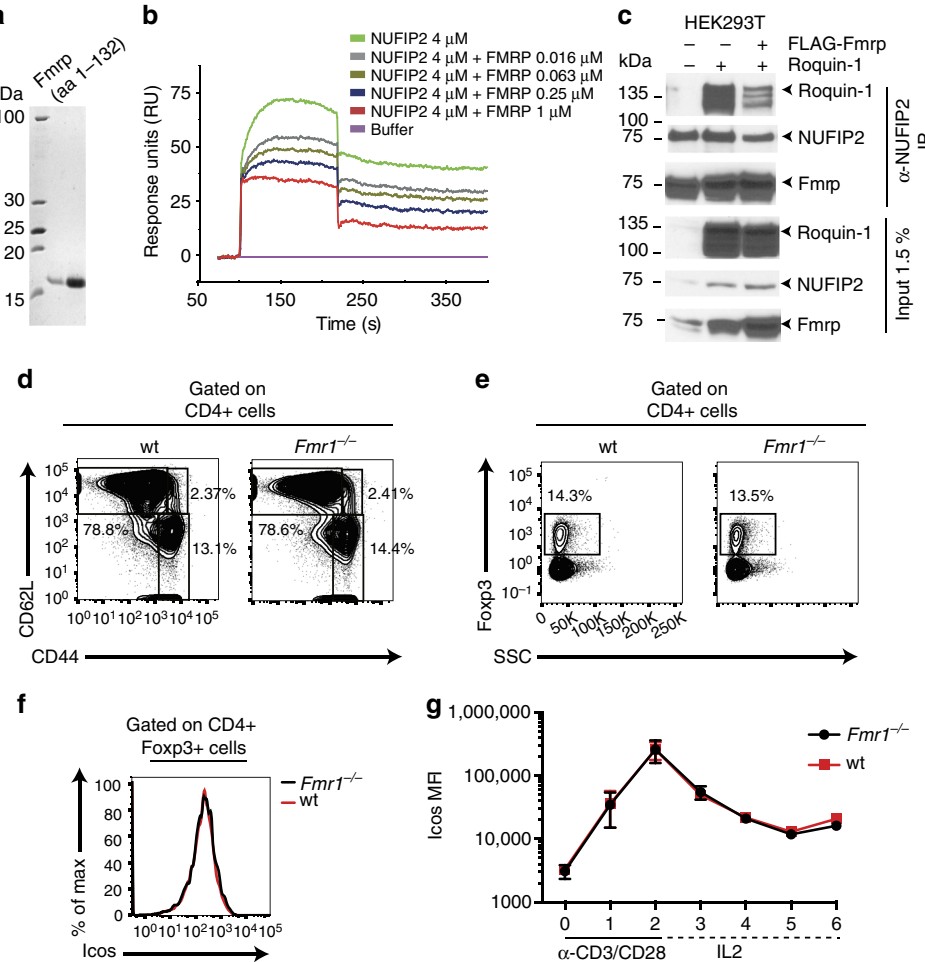

**Fig. 6** Mutually exclusive interaction of Nufip2 with Fmrp or Roquin. **a** Representative coomassie-stained PAGE of purified mFmrp (aa 1–132). **b** Biacore sensogram recording of the binding of NUFIP2 (aa 255–441) injected at 4 μM to surface-coupled Roquin-1 (aa 2–441) in the presence of increasing amounts (0.016–1 μM) of Fmrp (aa 1–132). The quality of Biacore experiments was assessed by two independent injections of 1 μM Fmrp and 4 μM NUFIP2 at different time points during each experiment. **c** Immunoprecipitation of endogenous NUFIP2 from lysates of HEK293T cells that were transiently transfected with Roquin-1 with or without co-transfection of FLAG-Fmrp using a monoclonal Nufip2 antibody. Immunoblot analysis to detect Nufip2, Fmrp and Roquin-1/2 after Nufip2 immunoprecipitation (IP) is shown. **d–g** Analysis of splenic CD4$^+$ T cells from wild-type (wt) and Fmrp-deficient (Fmr1$^{-/-}$) mice. **d, e** Flow cytometry analysis of CD4$^+$ T cells showing naive (CD62L$^{hi}$ CD44$^{lo}$), effector-memory (CD62L$^{lo}$ CD44$^{hi}$), or central-memory (CD62L$^{hi}$ CD44$^{hi}$) (**d**) or **e** Foxp3$^+$ Tregs among CD4$^+$ pre-gated T cells. **f** Flow cytometry analysis of ICOS expression on the surface of Foxp3$^+$ splenic CD4$^+$ T cells. **g** Quantified ICOS expression on CD4$^+$ T cells isolated from Fmrp-deficient or wild-type mice in a 6-day in vitro culture. One representative of two (**c**) or three (**b**) independent experiments are shown in **b** and **c**. In **d–f** data are representative of four mice per genotype and **g** shows mean and SD of four mice per genotype

present between nucleotides 2211 and 2271 of *ICOS* mRNA, a region that has already been implicated in Roquin-mediated regulation of *ICOS* via a miR-101 binding site[17,33]. However, our data demonstrated direct binding of Roquin to the *ICOS* 3′-UTR, and reporter analyses in Dicer- or Argonaute-deficient cells revealed Roquin-1-dependent repression of ICOS in an miRISC-independent manner[15].

Roquin has been shown to bind conserved tri-loop[2–4,6] or hexa-loop structures[1] in vitro. Additionally, crosslinking and immunoprecipitation experiments showed interaction with predicted U-rich stem-loops of different lengths in cells[7]. Therefore, we employed the LocARNA algorithm[39] for the identification of evolutionary conserved structured motifs within the mapped region of the *ICOS* 3′-UTR. Two conserved stem-loops, an octa-loop with a U-rich loop and a tri-loop without characteristics of the canonical CDE[6], were identified in the mapped region between nts 2211–2271 (Fig. 7b). To determine how Roquin recognizes the mapped *cis*-element we performed electrophoretic

mobility-shift assays (EMSAs) with recombinant Roquin-1 (aa 2–441) and in vitro-transcribed RNA. We extended the mapped sequence at the 5′ end by 28 nts to facilitate correct folding of the predicted octa-loop structure (*ICOS* nts 2183–2271). Roquin bound strongly to the mapped *cis*-element (nts 2183–2271) (Fig. 7c). The mode of interaction with this *cis*-element was further addressed by competition experiments. Efficient competition was only obtained through the unlabeled full *cis*-element (nts 2183–2271) RNA, whereas shorter RNA fragments were unable to affect the binding (Supplementary Fig. 6d). In line with this, Roquin bound only very weakly to shorter RNA fragments representing the full octa-loop containing element (Fig. 7d), the element containing only the 5′ part of the octa-loop (Fig. 7e) or the extended tri-loop containing element (Fig. 7f). Surprisingly, in contrast to its interaction with the CDE[3,6], Roquin did not bind to the isolated tri-loop-containing element (Fig. 7g). Nevertheless, additional experiments established that also this unconventional tri-loop can contribute to regulation. In reporter assays carried

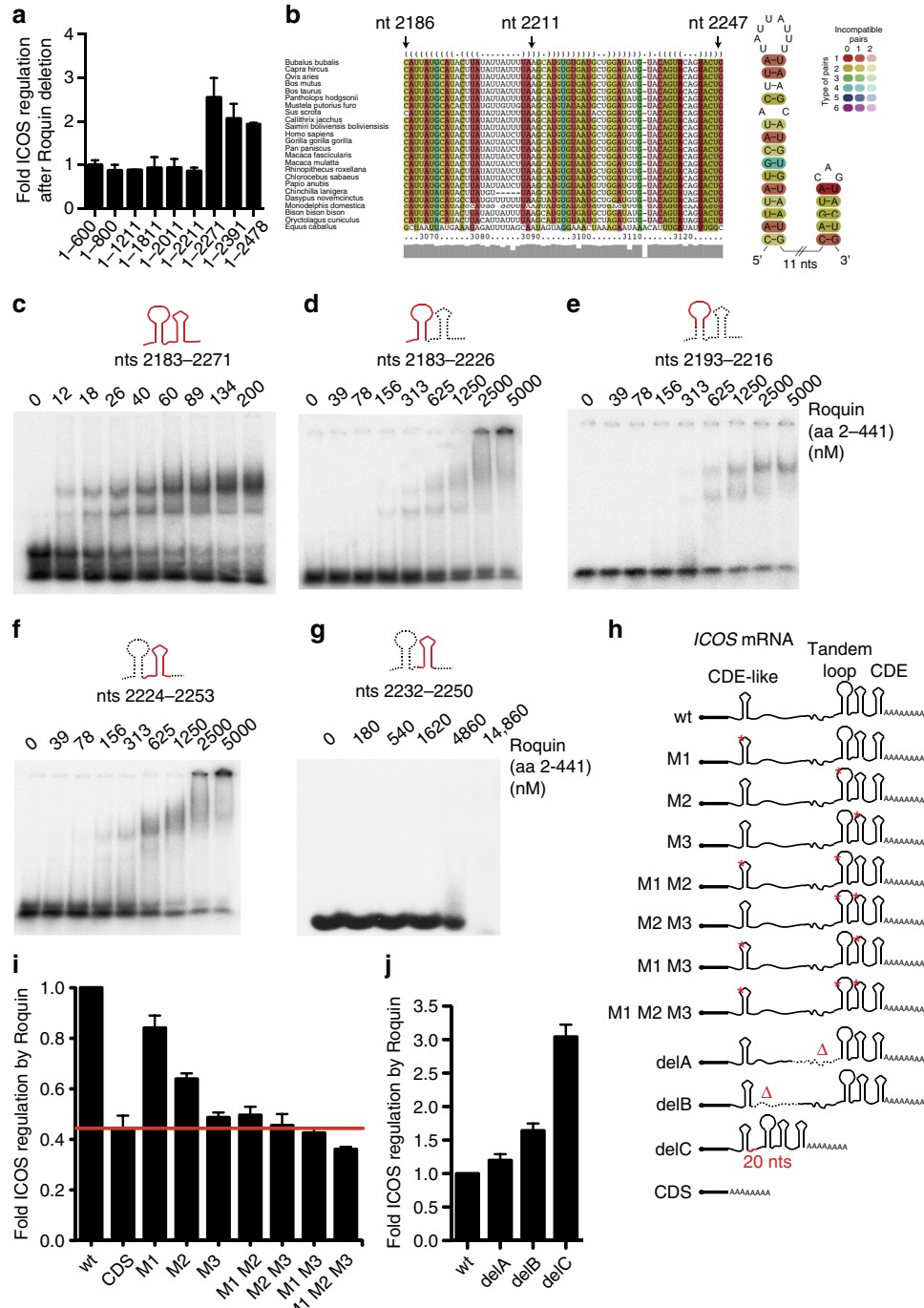

**Fig. 7** An unconventional *cis*-element in the *ICOS* mRNA is recognized by Roquin and critical for post-transcriptional repression. **a** Reporter regulation of gradually shortened *ICOS* 3'-UTR fragments in response to 4' OH-tamoxifen (4' OH-TAM) induced deletion of endogenous Roquin in *Rc3h1*^fl/fl; *Rc3h2*^fl/fl; *Cre-ERT2* MEF cells. ICOS expression was quantified by flow cytometry. Fold regulation was determined by dividing (ICOS MFI +4' OH-TAM)/(ICOS MFI –4' OH-TAM) and normalized to the regulation of ICOS 1–600 (CDS). **b** Sequence-structure alignment of the putative Roquin-active *ICOS cis*-element and consensus secondary structure of putative Roquin-binding stem-loop motifs are shown. The number of different types of base pairs for a consensus pair is indicated by different colors, the number of incompatible pairs by the saturation of the consensus base pair. A consensus secondary structure is depicted on the right. **c–g** Electrophoretic mobility-shift assays (EMSAs) with increasing amounts of Roquin-1 (aa 2–441) incubated with *ICOS* RNA fragments as indicated. Labeled in red is the part of the consensus secondary structure shown in **b** that is included in the respective *ICOS* RNA fragment. **g** At the highest protein concentration unspecific RNA binding was observed, as detected by a broad smear of retarded RNA. **h** Schematic representation of ICOS reporter constructs bearing mutant stem-loops. Mutations designed to destroy the secondary structure of selected RNA elements are indicated by asterisks. **i, j** Reporter regulation of the stem-loop and deletion mutants from **h** in response to 4' OH-TAM induced deletion of endogenous Roquin in *Rc3h1*^fl/fl; *Rc3h2*^fl/fl; *Cre-ERT2* MEF cells. Fold regulation was determined by dividing (ICOS MFI +4' OH-TAM)/(ICOS MFI –4' OH-TAM) and normalized to the regulation of ICOS wt. Error bars in **a**, **i** and **j** indicate mean and SDs of two (**a**) or four (**i, j**) independent experiments. In **c–g** representatives of three independent experiments are shown

out in Roquin-deficient MEF cells with doxycycline-inducible overexpression of Roquin-1, the *ICOS* 3′-UTR deletion construct that ended before the tandem cis-element (F2 in Supplementary Fig. 6e) only showed half of the reporter regulation as compared to full-length *ICOS* mRNA (wt) (Supplementary Fig. 6e). Additional deletion of a region encompassing the much further 5′ localized Roquin-bound CDE-like stem-loop (F3), which we have identified earlier by mapping the Roquin/*ICOS* mRNA interaction[11,40], attenuated the reporter regulation even more. On the other hand, neither the deletion of this CDE-like element alone affected reporter regulation (F1) nor did the combined deletion of the CDE-like tri-loop element together with the perfect consensus CDE at the very end of the 3′-UTR (F4). These data indicated a cooperative function of the CDE-like tri-loop element and the unconventional tandem *cis*-element located hundreds of nucleotides downstream. To confirm that all three stem-loops in both *cis*-elements contribute to Roquin-mediated ICOS repression in a cooperative manner, we tested ICOS reporters with single and combined double and triple point mutations in the loop-forming sequences (Fig. 7h, i). Mutating the loop of the CDE-like tri-loop (M1) or the octa-loop of the tandem element (M2) were much less effective compared to the mutation of the tri-loop in the tandem element (M3), which almost completely inhibited the regulation of the ICOS reporter by endogenous Roquin proteins (Fig. 7i). However, combining the mutations of CDE-like and octa-loop (M1 M2) impaired Roquin-mediated regulation to a similar extent, indicating the cooperative activity of all three stem-loops. This assumption received further support by the observed complete inhibition of regulation in all other combinations of two or more mutations (Fig. 7i compare M1 M2, M2 M3, M1 M3 or M1 M2 M3). Interestingly, the shortening or deletion of sequences between the CDE-like tri-loop and the tandem *cis*-element increased the regulation of the reporter by endogenous Roquin proteins (Fig. 7h, j).

Since NUFIP2 has been shown to interact with polyadenylated RNA[36], we tested whether NUFIP2 would bind directly to the mapped *ICOS* cis-element or to another tandem element present in the *Ox40* 3′-UTR that is required for the regulation of the Ox40 costimulatory receptor[1,3,40]. Specifically, we performed EMSAs with NUFIP2 (aa 255–411) and Roquin (aa 2–441) either with the *ICOS* RNA (nts 2183–2271) (Fig. 8a–c) or *Ox40* (nts 1064–1126) (Fig. 8d–f). Compared to Roquin (Fig. 8b), NUFIP2 bound the *ICOS* cis-element with much lower affinity (Fig. 8a), which was similarly true for the *Ox40* cis-element (Fig. 8d, e). We next investigated whether NUFIP2 interacted with the *cis*-element in the presence of Roquin. Intriguingly, addition of NUFIP2 caused a super-shift of Roquin-1-bound *ICOS* (nts 2183–2271) (Supplementary Fig. 7a), indicating the formation of a ternary complex. Quantifying the interaction of Roquin with these *cis*-elements in binary or NUFIP2-containing ternary complexes showed a 3–4-fold better binding in the presence of NUFIP2 (Fig. 8b, c, e, f and Supplementary Fig. 7b, c). These results support a model in which NUFIP2 increases the binding affinity of Roquin in a ternary complex with the mapped *ICOS* or *Ox40* cis-elements. Furthermore, the data explain how these unconventional *cis*-elements are recognized at low or endogenous levels of Roquin (Supplementary Fig. 7d).

## Discussion

The family of RNA-binding proteins has been extended enormously by recent findings showing that proteins without known RNA-binding domains interact with mRNA inside cells[28,41]. The current challenge therefore is to elucidate how such interactions contribute to specific post-transcriptional gene regulation.

Employing a targeted siRNA screening approach, we identified and characterized NUFIP2 as a cofactor of Roquin function. So far, little is known about the composition of the mRNP assembling around Roquin. It has been previously shown that the carboxy-terminus of Roquin-1 interacts with the deadenylase complex[5,6]. Furthermore, the amino-terminal part of Roquin-1 interacts with Ddx6/Rck and Edc4, two cofactors of mRNA decapping[15]. However, it is unclear whether these interactions are direct. In contrast, NUFIP2 not only co-immunoprecipitated with full-length and amino-terminal fragments of Roquin-1 from cell lysates, but also interacted as purified recombinant Roquin-1 and NUFIP2 protein fragments with high affinity ($K_D$ of <200 nM). NUFIP2 is therefore a direct binding partner of the amino-terminal part of Roquin. The recently reported direct interaction of NUFIP2 and Ddx6[37] may further enhance Roquin-dependent post-transcriptional repression. However, since recruitment of the carboxy-terminus of Roquin was sufficient to induce reporter mRNA degradation, this interaction may not be required[5].

A molecular property that connects Roquin and NUFIP2 is their shared localization in RNA granules. For some instances of such co-localization the biophysical basis has been demonstrated to lie in the aggregation of RNA-binding proteins containing LCR. They are enriched in polar amino acids like asparagine, glutamine, serine, tyrosine and glycine[42]. In fact such LCRs with enrichment in glutamine and asparagine can be found in peptide sequences of NUFIP2 as well as of Roquin[15,37]. We used a recently developed biochemical assay to precipitate proteins that have a tendency to aggregate with b-isoxazole[35] and also determined the localization of Roquin or NUFIP2 in Nufip2- or Roquin-deficient cells. In these experiments we show that both proteins aggregate, albeit with different b-isoxazole concentration requirements, and localize to stress granules independent of each other. Nevertheless, upon overexpression of Roquin, NUFIP2 becomes enriched in P bodies. Together these data reveal that the binary interaction between NUFIP2 and Roquin is not a driving force for heterotypic aggregation of both proteins, but may rather be involved in specific functions of both proteins.

Our work demonstrates the importance of NUFIP2 in Roquin-mediated *ICOS*-regulation. This target mRNA shows interesting and unconventional features: first, with 1978 nts the 3′-UTR of *ICOS* is remarkably long. The 3′-UTR contains multiple non-essential binding sites, at least one of which forms a perfect CDE. Furthermore, the mapped essential *cis*-element is composed of tandem stem-loops that are both required for efficient recognition by Roquin. Finally, this essential tandem stem-loop appears to depend functionally on additional binding of Roquin to another binding site being located in a long distance. We have observed before a cooperation of two binding sites in the *Ox40* 3′-UTR to equally and independently contribute to Roquin-induced mRNA decay[1]. It was therefore surprising to find that in the *ICOS* 3′-UTR the remotely 5′ located binding site enables the function of the essential 3′ binding site, while the sequences between both *cis*-elements appeared to inhibit post-transcriptional repression by Roquin. These findings suggest an additional level of regulation involving the secondary structure of this 3′-UTR, which may dynamically change the physical distance of both elements and therefore enable or reduce the effect of the bound *trans*-acting factors.

Roquin bound to the essential *cis*-element of the *ICOS* 3′-UTR with enhanced affinity in the presence of NUFIP2, and similar results were obtained for another Roquin-regulated tandem stem-loop containing *cis*-element from the *Ox40* 3′-UTR. A question that remains to be answered is how exactly NUFIP2 increases the binding affinity of Roquin to the mapped regions. One possibility is that NUFIP2 binding may promote Roquin to adopt a certain conformation thereby enabling additional or stronger contacts of

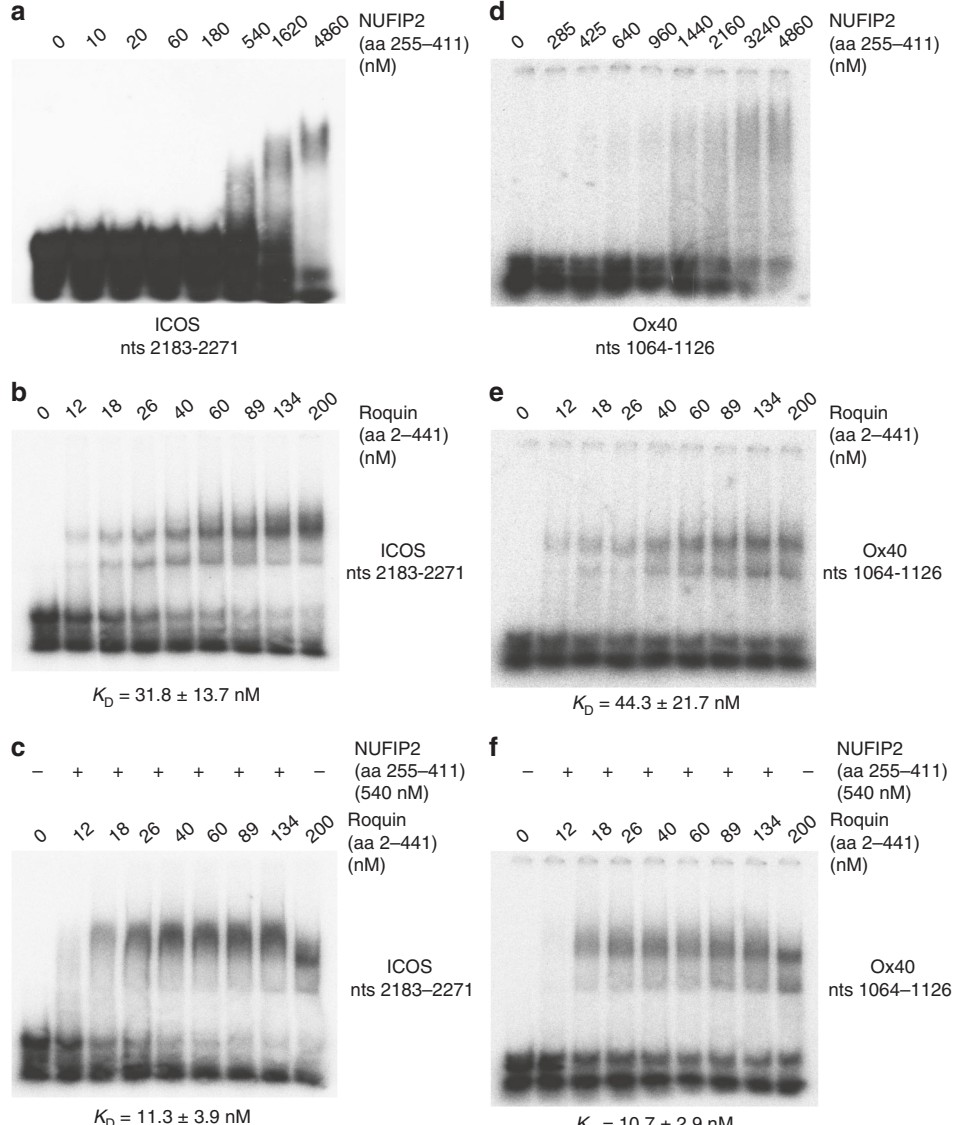

**Fig. 8** Cooperative binding of NUFIP2 and Roquin to response elements in *ICOS* and *Ox40* mRNA. **a** EMSA performed with the *ICOS* (nt 2183–2271) RNA (**a–c**) or *Ox40* (nt 1064–1126) RNA (**d–f**) using increasing amounts (0–4860 nM) of NUFIP2 (aa 255–411) (**a**, **d**) or increasing amounts (0–200 nM) of Roquin-1 (aa 2–441) (**b**, **e**), or using increasing amounts (0–200 nM) of Roquin-1 (aa 2–441) in the presence or absence of 540 nM NUFIP2 (**c**, **f**)

its RNA binding domain to specific RNA elements (Supplementary Fig. 7d). Such a mechanism of cofactor-dependent changes in binding affinity has recently been shown for the yeast DEAD box protein Rok1, which interacts with a dsRNA duplex in the pre-rRNA between 5.8 S and 18 S rRNAs. Target specificity of Rok1 was established by binding to its cofactor Rrp5. Rrp5 binding induced changes in the Rok1 secondary structure, thereby enhancing Rok1 target specificity without binding to RNA itself [43].

On the other hand, NUFIP2 was originally described as an RNA-binding protein recognizing poly(G) homopolymers[36]. It might therefore bind RNA sequence or structure elements in the *ICOS* 3′-UTR by itself. Considering the mapped *cis*-element and the higher affinity of Roquin for this element, it is more likely that Roquin recruits NUFIP2 to this specific RNA element than vice versa. In this scenario NUFIP2 contributes, in a complex with Roquin, additional contacts to the recognized *cis*-element and thereby enhances or alters sequence and shape-specific RNA recognition by Roquin (Fig. 8c, f). Very recently a similar scenario has been described for *ASH1* mRNA recognition by a motor-

transport complex in budding yeast[44]. In the nucleus She2p recognizes *cis*-elements in the RNA with modest affinity and specificity. After nuclear export, the RNA-binding protein and myosin adapter She3p joins the complex, thereby improving affinity and specificity for *ASH1* mRNA. A crystal structure of the ternary yeast complex revealed that She3p improves binding by joining the complex as an unfolded protein, contacting both the globular She2p protein and the RNA[44]. In analogy, also the RNA-binding ROQ domain of Roquin is globular, whereas the interacting NUFIP2 fragment is largely unfolded[45] and modulates the RNA binding. Given the large number of recently discovered unfolded RNA-binding proteins[28,41,46], it is likely that more examples are to be discovered, where unfolded proteins such as NUFIP2 modulate the RNA binding and function of globular proteins like Roquin. Our study provides evidence that also the combination of different RNA *cis*-elements contribute to the complexity and precision of post-transcriptional regulation.

Future experiments should address whether the physical and functional cooperation of Roquin and NUFIP2 is specific for certain *cis*-elements and target mRNAs and whether NUFIP2 also

increases the affinity and adapts the specificity to distinct *cis*-elements in target mRNAs of FMRP. Regardless of these open questions, our findings contribute to the emerging view that post-transcriptional control appears to rely on a complex code of *cis*- and *trans*-acting elements, reminiscent of what has been shown for transcriptional control.

## Methods

**Cloning**. ICOS reporter constructs consisting of human *ICOS* coding sequence followed by fragments of the *ICOS* 3′-UTR or the *TNF* CDE$_{260}$ have been described previously[8,15]. Expression constructs for Roquin-1, Roquin-1$^{san}$, GFP-Roquin-1 full-length, GFP-Roquin-1 (aa 1–509), GFP-Roquin-1 (aa 509–1130), GFP, and Regnase-1 have been described previously[8,15]. For generation of Roquin-1-P2A-mCherry, a mammalian codon-optimized sequence encoding an SGSG-linker followed by P2A-mCherry was synthesized by Entechelon GmbH (now Eurofins Genomics) and cloned behind the coding sequence of Roquin-1 via *Not*I restriction sites. Human NUFIP2 was amplified from cDNA clone IMAGE:40128910 obtained from Source BioScience LifeSciences. NUFIP2, NUFIP2 (aa 255–411), NUFIP2 (aa 411–695), and mouse Fmrp (aa 1–132) were amplified from cDNA using forward and reverse primers. Primer sequences are listed in Supplementary Table 1. To generate GFP-NUFIP2, NUFIP2 cDNA was cloned downstream of GFP as described previously[47]. siRNA-resistant GFP-NUFIP2, stem-loop mutants (M1–M3), or deletion constructs (delA, delB, delC) of the *ICOS* 3′-UTR reporter were obtained by introducing targeted mutations into plasmid DNA using the Quik-Change® II XL Site-Directed Mutagenesis Kit (Agilent) according to the manufacturer's instructions. Site-directed mutagenesis was also employed for generation of the N-terminal NUFIP2 protein fragments (NUFIP2 aa 1–255 and aa 1–411) by introducing STOP codons at defined positions within the cDNA. Primer sequences are listed in Supplementary Tables 1 and 3. cDNAs were cloned into retroviral expression vectors (KMV IRES GFP, MSCV IRES Thy1.1 or pRetroX-Tight) or transient transfection vectors (pDest12, pLNCX2) as described before[1,3,11,15]. For doxycycline-inducible lentiviral gene expression, sequences encoding Roquin-1 or Roquin-1-P2A-mCherry were inserted into plentiCMV$_{tight}$ Neo DEST (Addgene plasmid #26432). plentiCMV$_{tight}$ Neo DEST, the rtTA3-encoding vector plenti CMV rtTA3 Blast (Addgene plasmid #26432) and plenti PGK Hygro DEST for stable lentiviral gene expression (Addgene plasmids #19066) were donated by Dr. Eric Campeau[48].

**Expression and purification of recombinant proteins**. Expression of Roquin-1 (aa 2–441) has been described previously[1,3]. For expression of NUFIP2 (aa 255–411), the fragment was inserted into the pOPINS3C and expressed as N-terminal His$_6$-SUMO-fusion protein together with a C-terminal His$_6$ tag in *Escherichia coli* BL21(DE3)pRIL. Cells were grown at 37 °C in LB medium with 100 μg/mL ampicillin and 34 μg/mL chloramphenicol. At OD$_{600}$ = 0.7, the cell cultures were induced by adding 0.5 mM isopropyl β-D-1-thiogalactopyrano-side (IPTG) and overnight growing conditions changed to 20 °C. Next, the cells were harvested and resuspended in lysis buffer (500 mM NaCl, 20 mM imidazole, 0.1% Tween20, 10 μg/mL DNase I, protease inhibitors, 50 mM 4-(2-hydroxyethyl)-1-piperazineethanesulfonic acid (HEPES), pH 7.5) and sonicated at 4 °C. After centrifugation, the cleared lysates were applied to a HisTrap FF column (GE Healthcare). The His$_6$-SUMO tag was cleaved off by PreScission protease digestion at 4 °C overnight and removed using the ion exchange columns HiTrap Q FF and HiTrap SP FF (GE Healthcare). Protein bound to the HiTrap SP FF column was concentrated and further purified using a Superdex 75 10/300 GL column (Amersham Pharmacia Biosciences) in 200 mM NaCl and 50 mM HEPES pH 7.5 buffer.

Fmrp (aa 1–132) was inserted into the pOPINJ and expressed as N-terminal His$_6$-GST-fusion protein in *E. coli* BL21Star(DE3). Cells were grown at 37 °C in LB medium with 100 μg/mL ampicillin until they reached OD$_{600}$ = 0.7. Then 0.5 mM IPTG was induced to the cell cultures and left to grow overnight at 20 °C. In the morning the cells were harvested, resuspended in lysis buffer (500 mM NaCl, 20 mM imidazole, 0.1 % Tween20, 10 μg/mL DNase I, protease inhibitors, 50 mM HEPES, pH 7.5) and sonicated at 4 °C. The lysates were centrifuged and the supernatant was applied to a HisTrap FF column (GE Healthcare). His$_6$-GST tag was cut off by the PreScission Protease at 4 °C overnight and then removed using a GSTTrap FF column (GE Healthcare) and GST beads (Glutathione Sepharose High Performance; GE Healthcare). The unbound protein was concentrated and gel filtrated using a HiLoad 16/600 Superdex 75 pg (Amersham Pharmacia Biosciences) in 250 mM NaCl and 10 mM HEPES pH 7.5. The different purified proteins were ≥95% pure with OD 260/280 values below 0.65 indicating that the proteins were essentially free of nucleic-acid contaminations.

**Surface plasmon resonance**. A BIACORE 3000 instrument (Biacore Inc.) was used to analyze Roquin-NUFIP2 binding. Roquin (aa 2–441) was immobilized on the CM5 sensor chip with amine coupling (Biacore Inc.) at a concentration of 50 μg/mL in 10 mM sodium-phosphate buffer pH 5.7. NUFIP2 (aa 255–411) was injected onto the sensor chip using the concentrations 0.063, 0.125, 0.25, 0.5, 1, 2, and 4 μM at 30 μL/min flowrate and at 20 °C in running buffer (150 mM NaCl,

0.05% Tween20, 10 mM HEPES). When NUFIP2 (aa 255–411) and FMRP (aa 1–132) were injected together, the concentration of NUFIP2 was 4 μM and the concentrations of FMRP were 0.016, 0.063, 0.25, and 1 μM in the running buffer (250 mM NaCl, 0.05% Tween20, 10 mM HEPES). The equilibrium dissociation constant $K_D$ was calculated from steady-state measurements using the BIAeva-luation program (Biacore Inc.).

**RNAs**. The *ICOS* (nt 2232–2250) RNA fragment was purchased from IBA GmbH (Göttingen, Germany), where it was synthesized and purified via PAGE (poly-acrylamide gel electrophoresis) followed by two steps of desalting.

The *ICOS* (nt 2183–2271), (nt 2193–2216), (nt 2224–2253), (nt 2251–2271), (nt 2183–2226), and Ox40 (nt 1064–1126) RNA fragments were in vitro-transcribed using the MEGAshortscript™ Kit (Ambion). Primers with the T7 promoter sequence at the 5′ end that were used as templates for in vitro transcription are listed in Supplementary Table 3. After transcription, the RNA was purified based on manufacturer's instructions, starting with DNase digestion, proceeding with phenolic extraction, followed by RNA precipitation using ethanol. The integrity of the RNA was confirmed by native or denaturing urea PAGE.

The library of siGENOME siRNA pools was custom-made. Individual siRNA duplexes targeting STAU1 (D-011894-01, D-011894-02, D-011894-03, D-011894-04), CNOT1 (D-015369-01, D-015369-02, D-015369-03, D-015369-04) and NUFIP2 (D-021280-01, D-021280-03, D-021280-04, D-021280-17) as well as Roquin-targeting siRNA pools (M-044230-01) and non-targeting control siRNAs (D-001210-01) were from GE Healthcare (Dharmacon, Lafayette, CO, USA), in addition, published siRNA sequences[9] against RC3H1 (5′-GAUCGAGAGUUACUAUCCAdTdT-3′) or RC3H2 (5′-GCUUGAAAAGUAUCGAUUAdTdT-3′) were custom synthesized (Invitrogen).

**Electrophoretic mobility-shift assay**. RNA fragments for electromobility shift assays (EMSAs) were radioactively labeled using T4 polynucleotide kinase and [γ$^{32}$P]ATP. Sepharose spin columns (NucAway; Ambion) were used to separate the RNA from free nucleotides. The radioactively labeled RNA (5 nM), the Roquin-1 protein (aa 2–441) and the tRNA competitor (30 μg/mL) were incubated in Hepes/NaCl/MgCl$_2$ (HNM) buffer (10 mM HEPES pH 7.5, 150 mM NaCl, 2 mM MgCl$_2$) and 4% glycerol in the final volume of 20 μL for 30 min at 25 °C. In competition experiments, unlabeled competitor RNA was added afterwards and incubated for another 15 min at 25 °C. The samples were resolved by native Tris/Borate/EDTA (TBE) PAGE (4% polyacrylamide, 1× TBE buffer). Gels were ana-lyzed using radiograph films or Fuji Imaging plates exposed in the FLA-3000, after 15 min incubation in fixing solution (30% (vol/vol) methanol, 10% (vol/vol) acetic acid) and vacuum drying. The signal was quantified using ImageJ (version 1.51). The curve fitting and the calculation of the $K_D$s was done using the software Origin (version 9.0.0G).

**Mice and cell culture**. The mouse lines *Rc3h1*$^{fl/fl}$; *Rc3h2*$^{fl/fl}$; CAG-CAR$^{STOP-fl}$; CreERT2 (denoted *Rc3h1*$^{fl/fl}$; *Rc3h2*$^{fl/fl}$; *Cre-ERT2), Rc3h1*$^{fl/fl}$; *Rc3h2*$^{fl/fl}$; *Cd4-Cre-ERT2; rtTA* and *Rc3h1*$^{fl/fl}$; *Rc3h2*$^{fl/fl}$; *Cd4-Cre* (denoted *Rc3h1/2*$^{fl/fl}$; *Cd4-Cre*) and *Fmr1*$^{−/−}$ mice have been described previously[1,3,11,49]. All animals were housed in a pathogen-free barrier facility in accordance with the Helmholtz Zentrum München, the Ludwig-Maximilians-University München, and Erasmus MC insti-tutional, state, and federal guidelines. HEK293T cells, *Rc3h1/2*$^{−/−}$ mouse embryonic fibroblast (MEF) cells, and *Zc3h12a*$^{−/−}$MEF cells have been described before[8,15]. MEF, HEK293T, and HeLa cells were cultured in a cell culture incubator at 37 °C, 10% CO$_2$ in DMEM (ThermoFisher Scientific) supplemented with 10% (v/v) fetal bovine serum (FBS) (PAN BIOTECH), penicillin-streptomycin (100 U/mL) (ThermoFisher Scientific) and 10 mM HEPES, pH 7.4 (ThermoFisher Scientific). Primary mouse T cells were cultured at 37 °C, 10% CO$_2$ in DMEM supplemented with 10% (v/v) FBS, 10 mM HEPES pH7.4, penicillin-streptomycin (100 U/mL), non-essential amino acids (ThermoFisher Scientific), and 50 μM β-mercaptoethanol (Sigma-Aldrich).

Mouse CD4$^+$ T cells were isolated from spleen and lymph nodes of C57BL/6 mice with the EasySep™ Mouse CD4$^+$ T-cell Isolation Kit (StemCell Technologies). For activation, T cells were stimulated with anti-CD3 (0.25 μg/mL) and anti-CD28 (2.5 μg/mL) on six-well plates coated with goat-anti-hamster immunoglobulin G (IgG) (0.05 mg/mL in phosphate buffered saline (PBS)) for 40 h and skewed towards Th1 differentiation by addition of IL-12 (10 ng/mL) and anti-IL-4 (10 μg/mL). After stimulation, cells were expanded in complete media containing 20 U/mL of recombinant human IL-2 (ProleukinS, Novartis) for 2–4 days.

For cytokine stimulation, cells were kept in complete media without IL-2 24 h prior stimulation with anti-CD3 (0.25 μg/mL) in combination with anti-CD28 (2.5 μg/mL) or anti-ICOS (2.5 μg/mL) for 12 h. Alternatively, cells were incubated in media containing IL-2 (20 U/mL), IL-6 (5 ng/mL), IL-10 (50 ng/mL) or anti-CD3 and anti-CD28 for 20 h.

**Antibodies and reagents**. Monoclonal antibodies against Roquin-1 and Roquin-2 (3F12, 18F8, Q-4-2), CD3 (145-2C11), and CD28 (37N) have been described before[8,11,15]. To generate monoclonal antibodies against NUFIP2, a Wistar RjHan: WI rat was immunized with purified GST-tagged full-length human NUFIP2

protein using standard procedures as described[50]. The hybridoma cells of NUFIP2-reactive supernatants were cloned twice by limiting dilution. Experiments in this study were performed with NUFIP2 23G8 (rat IgG2b/κ) and 14G9 (rat IgG2a/κ). Antibodies against Tubulin (B-5-1-2), G3bp1 (sc-81940), and p70 S6 kinase (p70S6K, sc-8418) were purchased from Santa Cruz. Anti-CNOT1 (14276-1-AP) was from Proteintech. Polyclonal anti-NUFIP2 (A301–600A) and anti-Roquin-1 (A300–514A) antibodies were obtained from BETHYL Laboratories. Anti-GFP (A-6455) was from ThermoFisher. Anti-GAPDH (6C5) was from Calbiochem. Anti-Fmrp (#4317), anti-Phosho-Stat3 (#9145) and anti-Phosho-Stat5 (#4322) were obtained from Cell Signaling. Flow cytometry antibodies against human ICOS (anti-CD278, clone ISA-3), mouse CD4 (Rm4-5), mouse CD62L (Mel-14), mouse CD44 (IM-7), mouse Foxp3 (FJK-16s), and mouse ICOS (7E.17G9) were purchased from eBioscience. Amine-reactive fluorescent dyes Alexa Fluor® 488 Carboxylic Acid Succinimidyl Ester and Pacific Blue™ Succinimidyl Ester were obtained from Life Technologies. 4′ OH-tamoxifen, Doxycycline-Hyclat, and b-isox were obtained from Sigma-Aldrich.

**Transfection, transduction, and generation of stable cell lines**. Retro- and lentiviral supernatants were produced by calcium-phosphate transfection of HEK293T as described previously[11,15]. For virus transduction, 100,000 MEF or HeLa cells were plated onto each well of a six-well plate 18 h prior to transduction. Infection with the desired amount of viral supernatant (100 μL to 2.5 mL) was performed by centrifugation for 2 h at 32 °C, 300×g in the presence of 8 μg/mL polybrene.

HeLa reporter cells were generated by co-transduction of HeLa cells with plentiCMV_tight Roquin-P2A-mCherry, the reverse transactivator (rtTA3), and plenti PGK ICOS. Starting 48 h after transduction, cells were cultured in medium supplemented with 500 μg/mL Geneticin (Neomycin) (Gibco), 2.85 μg/mL Blasticidin S HCl (Gibco) and 500 μg/mL Hygromycin B (Invitrogen) for 7 days to select for stably infected cells. A cell line with stable ICOS and dox-inducible co-expression of Roquin and mCherry was obtained by single cell cloning. A stable Rc3h1/2−/− MEF cell line allowing doxycycline-inducible co-expression of Roquin-1 and mCherry was generated similarly by transduction with plentiCMV_tight Roquin-P2A-mCherry and the reverse transactivator (rtTA3). NUFIP2–deficient MEFs and HeLa cells were generated by transduction with lentiCRISPR vectors encoding the respective NUFIP2-targeting sgRNAs together with Cas9. Starting 48 h after transduction, cells were cultured in the presence of 2 μg/mL puromycin (Sigma) for 3 days to select for infected cells. After single cell cloning, the clone showing the lowest NUFIP2 expression, as verified by immunoblotting, was chosen as a stable NUFIP2 knockout cell line. Rc3h1/2−/− and Nufip2−/− MEF cells with doxycycline-inducible Roquin-1 overexpression were generated by co-transduction with plentiCMV_tight Roquin and the reverse transactivator (rtTA3) as described above. Nufip2-targeting sgRNAs (mNufip2#4: 5′-GGCTTGTTTCTACAACTTGC-3′, hNUFIP2: 5′-GCATGTTTCAGCGGCCTTTGGC-3′) were designed with the optimized CRISPR tool (http://crispr.mit.edu) provided by the Zhang lab and cloned into the plentiCRISPR EFS GFP via BsmBI restriction sites. plentiCRISPR EFS GFP was a kind gift from Dr. Benjamin Ebert (Addgene plasmid #57818)[51] and provides stable expression of GFP, sgRNA, and Cas9 nuclease from one vector.

**Functional assays**. Rc3h1/2−/− MEFs with doxycycline-inducible Roquin-1-P2A-mCherry expression (described above) were employed for functional assays upon retroviral transduction with different ICOS reporters. Starting 48 h after transduction, cells were cultured in the presence of 1 μg/mL doxycycline to induce Roquin-1 and mCherry expression. ICOS expression was quantified 48 h post-infection by flow cytometry as described previously[3,11,15]. Similarly, Rc3h1fl/fl; Rc3h2fl/fl; Cre-ERT2 MEF cells were transduced with different ICOS reporters by retroviral infection. For in vitro deletion of Rc3h1 and Rc3h2, cells were cultured in the presence of 0.33 μM 4′-OH-tamoxifen for 4 days starting 48 h after transduction, followed by quantification of ICOS expression by flow cytometry as described before.

For colocalization experiments, Rc3h1/2−/− MEF cells with doxycycline-inducible Roquin-1 expression were transduced with GFP-Nufip2 by retroviral infection. Rc3h1/2−/− MEF cells lacking the cassette for doxycycline-inducible Roquin-1 expression were transduced with GFP-NUFIP2 for studying Nufip2 localization in the absence of endogenous Roquin expression.

Retroviral infection of Rc3h1fl/fl; Rc3h2fl/fl; Cd4-Cre-ERT2; rtTA CD4+ T cells with pRetroXtight GFP-NUFIP2 was performed as described previously[1]. For in vitro deletion of Rc3h1 and Rc3h2, T cells were treated with 1 μM 4′-OH-tamoxifen for 24 h prior to activation.

Reverse transfection of HeLa cells with siRNAs was performed with the lipid-based HiPerFect transfection reagent (Qiagen) according to the manufacturer's instructions.

For the analysis of mRNA stability transcription was stopped by addition of actinomycin D (5 μg/mL) to the cell medium. Cells were harvested at different time points after actinomycin D addition followed by total RNA isolation and reverse transcription and quantitative PCR (RT-qPCR)-based determination of mRNA amounts.

**Super-resolution microscopy (3D-SIM)**. For fluorescence microscopy, cells were seeded onto coverslips and treated with 1 μg/mL doxycycline for 14 h to induce Roquin overexpression or left untreated. Before fixation, cells were treated with 0.5 mM sodium arsenite (Sigma-Aldrich) for 1 h at 37 °C or left untreated. Cells were fixed with 2% formalin (ROTH) in PBS for 10 min and after a washing step in PBS containing 0.02% Tween20 (PBST) cells were permeabilized in PBS containing 0.02% Tween20 (ROTH) and 0.5 % Triton X-100 (AppliChem) for 10 min. Coverslips were treated with 2% bovine serum albumin (BSA) in TBST for 1–2 h at room temperature. Subsequently, cells were stained with the primary antibodies against Roquin (18F8), and either p70 S6 Kinase (1:1000 dilution) or G3bp1 (see above) followed by a second antibody staining using anti-rat Alexa Fluor 594 (Life Technologies) and anti-mouse CF405S (Biotium), respectively. GFP-NUFIP2 was stained with GFP-Booster ATTO488 (ChromoTek), while endogenous Nufip2 was stained with a primary antibody against Nufip2 (14G9) followed by a second antibody staining using anti-rat Alexa Fluor 594 (Life Technologies). Primary monoclonal antibody supernatants and the commercial G3bp1 antibody were diluted 1:100, while the secondary antibodies anti-rat Alexa Fluor 594 and anti-mouse CF405S were diluted 1:500 and 1:200, respectively. After a final fixation step in 4% formalin (ROTH) in PBS the cells were mounted in Vectashield (Vector Laboratories). Nuclei were stained with DAPI (Sigma-Aldrich). Images were acquired with a DeltaVision OMX V3 microscope. 3D SIM raw data were reconstructed and roughly corrected for color shifts with the software softWoRx 6.0 Beta 19 (unreleased). After establishing composite TIFF stacks with a custom-made macro in Fiji, the data were subsequently aligned again.

**Co-immunoprecipitation and immunoblotting**. For preparation of the protein lysates from CD4+ T cells, MEF, HeLa or HEK293T cells, cells were washed twice with cold PBS and lysed in 20 mM Tris-HCl, pH 7.5, 150 mM NaCl, 0.25% (v/v) Nonidet-P40, 1.5 mM MgCl₂, 1 mM dithiothreitol (DTT) supplemented with EDTA-free protease inhibitor mix (Roche) for 20 min on ice. Lysates were cleared by centrifugation at 10,000×g for 10 min at 4 °C and the protein concentration was determined with the Bio Rad protein assay (Bradford assay). Equal amounts of total protein (usually 50 μg protein/slot for detection of endogenous proteins) were separated by SDS-PAGE, transferred to a nitrocellulose membrane and analyzed by using primary antibodies (identified above) followed by horseradish peroxidase-conjugated secondary antibodies. Primary monoclonal antibody supernatants were diluted 1:10, while purchased antibodies were diluted according to the manufacturer's protocol. Secondary species-specific antibodies were used in a 1:3000 dilution. For protein detection, the Amersham ECL Prime Western Blotting Detection Reagent (GE Healthcare) and X-ray films were used.

For immunoprecipitation, antibody-coupled beads were incubated with 2–5 mg of protein lysate for 4 h at 4 °C in the presence or absence of RNAse (Roche). After washing three times with lysis buffer the beads were resuspended in Laemmli SDS sample buffer and boiled for 5 min at 95 °C for protein elution. The supernatant was subjected to immunoblotting. For GFP-immunoprecipitation from HEK293T cells (Fig. 5a, b), 5 μg polyclonal anti-GFP antibody was coupled to Protein G Dynabeads (ThermoFisher Scientific). Pulldown of endogenous Roquin-1/2 from MEF cells (Fig. 5c) was performed with anti-Roquin-1/2 (Q4-2)-coupled tosylactivated MyOne Dynabeads (ThermoFisher Scientific). Antibody coupling was performed according to the manufacturer's instructions. Roquin-1 was immunoprecipitated from HEK293T cells (Fig. 5e) with 4–5 μg polyclonal anti-Roquin-1 antibodies coupled to Protein A Dynabeads (ThermoFisher Scientific). Pulldown of endogenous NUFIP2 from HEK293T cells (Figs. 5d and 6c) was performed with 1 mL monoclonal anti-Nufip2 (23G8) supernatant coupled to Protein G Dynabeads. Immunoblot analysis of Roquin-1 and Roquin-2 was performed with monoclonal 3F12 antibody, while detection of Roquin-1 alone was performed with polyclonal Roquin-1 antibodies. Immunoblot analysis of Nufip2 expression was initially performed with the commercial polyclonal Nufip2 antibody (Figs. 3c, d and 5a, b) and in all other experiments with the self-made monoclonal antibody 23G8.

Pulldown of RNA-granule-like structures with b-isox was performed according to Kato et al.[35]. Briefly, MEF cells were lysed in 20 mM Tris buffer with 150 mM NaCl, 5 mM MgCl₂, 20 mM β-mercaptoethanol, 0.5% NP-40, 10% glycerol, EDTA-free protease inhibitor cocktail (Roche), 0.1 mM phenylmethylsulfonyl fluoride, 1:100 RNasin ribonuclease inhibitor (Promega, Madison, WI, USA), and 2 mM vanadyl ribonucleoside vanadyl complex (NEB, Ipswich, MA, USA). Lysates were incubated with 1–100 μM b-isox for 1 h at 4 °C on a rotating wheel. Reactions were centrifuged for 15 min at 14,000×g, 4 °C, resulting in the separation of b-isox-pelleted proteins and proteins in the supernatant. Supernatants were directly boiled in Laemmli SDS sample buffer, while b-isox pellets were washed twice with lysis buffer, resuspended in Laemmli buffer, boiled and loaded onto an SDS gel together with the supernatants for protein analysis.

**Quantitative RT-PCR**. RNA was isolated from cells with TRI reagent® (Ambion) or with the NucleoSpin® RNA Kit (Macherey-Nagel) according to the manufacturer's instructions and reverse transcribed into cDNA with the QuantiTect® Reverse Transcription Kit (Qiagen). Following reverse transcription, quantitative PCR assays (qPCR) were run on a Light Cycler 480 II machine using the Roche Universal Probe Library. Primers and universal probes are listed in Supplementary Table 4. Relative gene expression was determined with the Light Cycler 480 SW 1.5.1 software, and normalized to the expression of the reference gene Ywhaz.

**High-throughput siRNA screen**. High-throughput siRNA screening in HeLa cells was performed by reverse transfection of siRNAs in 96-wells using the HiPerFect transfection reagent (Qiagen). Forty microliters of the pre-diluted HiPerFect transfection reagent were dispensed manually into each well of a 96-well U-bottom plate (plate A) using a multichannel pipette. 8 μL of 1 mM siRNA library stocks and 32 μL of $H_2O$ were subsequently added by a robot, and after mixing, the robot dispensed half of the solution (40 μL) into a replicate plate (plate B). Eight microliters of 1 mM control siRNAs in 32 μL $H_2O$ (siRoquin, siCtrl) had been added manually before addition of the library stocks. Following an incubation time of 10 min to allow complex formation, HeLa reporter cells were added in complete DMEM medium to each well. Plates were incubated for 48 h and subsequently the medium was replaced by fresh medium containing 1 μg/mL doxycycline. After 18 h the cells were subjected to fluorescent cell barcoding[52,53] and antibody staining. Fluorescent cell barcoding was in principal performed as described in ref.[52]; however, the protocol was adapted to label cell surface proteins only. HeLa cells were washed once with PBS, trypsinized and transferred to U-bottom 96-well plates. Washing steps were now carried out by resuspension, centrifugation at 400×g and 4 °C for 5 min followed by aspiration of the supernatant with a vacuum pump containing an eight-channel adaptor. Cells were washed once with PBS, while dilutions of the amine-reactive fluorescent dyes Pacific blue Succinimidyl Ester and Alexa Fluor 488 Succinimidyl Ester were simultaneously prepared from 5 mg/mL stock solutions in PBS. Four different barcoding solutions were generated by employing both dyes in either a low (0.2 μg/mL) or a high (4 μg/mL) concentration. Cell pellets of each 96-well plate were resuspended in 50 μL of one of the four distinct barcoding solutions, and plates were sealed, vortexed, and incubated at 4 °C for 30 min. After washing twice with PBS+ 2% FCS, four plates with differential barcoding signatures were pooled into one. After spinning for 5 min at 400×g, 4 °C, cells were subjected to antibody staining as described below and analyzed by flow cytometry.

**Statistical analysis**. ICOS mean fluorescence intensity was determined with FlowJo Version 10.2. Statistical analysis was performed with GraphPad Prism 5.0d, and P-values were calculated with the Kruskal–Wallis test followed by Dunn's multiple comparison test. ICOS mRNA half-life values were calculated with GraphPad Prism by one-phase decay.

The Z′ factor in pilot experiments was calculated according to Zhang et al.[27]. All screen data were normalized into Z scores using the following equation:

$$Z = \frac{\text{ICOS MFI}_{sample} - \text{ICOS MFI}_{plate\ average}}{\text{SD plate}}$$

**Flow cytometry**. Antibody staining for flow cytometry was performed as described previously[8,11,15]. Briefly, cells were subjected to a 1:200 dilution of primary antibodies for 20 min at 4 °C, washed with PBS +2% FCS (FACS buffer) and subjected to a secondary antibody staining if necessary. After two final washing steps, cells were resuspended in 100 μL FACS buffer and acquired on a LSR II Fortessa cell analyzer (BD Bioscience).

**Sequence and structure conservation analysis**. All mRNA and sequence data used in this study were acquired from the NCBI Reference Sequence Database (RefSeq) collection release 69[54]. A data set was prepared by extracting all mRNAs of ICOS. Only mRNAs with a completely annotated 3′-UTR were used. The data set included 24 mammals. A complete list of species and associated identification numbers used in this study is given as Supplementary Table 5.

A combined multiple sequence-structure alignment of the data set sequences were obtained applying LocARNA version 1.8.1[39] with the option local-progressive followed by RNAalifold[55] from the ViennaRNA package version 2.1.8-2[56].

**Data availability**. The data sets generated during and/or analyzed during the current study are available from the corresponding author on reasonable request.

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

## Acknowledgements

We thank Claudia Lohs and Desiree Argiriu for expert technical assistance. For the provision of mice we would like to thank Thorsten Buch (*Cd4-Cre-ERT2*), Marc Schmidt-Supprian (*Rc3h1*fl), Wolfgang Wurst (*Rc3h2*fl) and Christopher Wilson (*Cd4-Cre*). We are grateful to Anjana Rao for supporting the screening approach in this project and like to thank Elke Glasmacher and Kai Höfig for critical reading of the manuscript. The work was supported by grants from the German Research Foundation including SFB 1243 project A01 to H.L., SPP-1935 to V.H. and D.N., SFB-1054 projects A03 and Z02 to V.H., HE3359/4–1 to V.H., the Initiative and Networking Fund of the Helmholtz Association (VH-NG738) to J.H. and from the European Research Council through an ERC-StG to V.H. and grants from the Fritz Thyssen and Else-Fresenius-Kröner foundations to V.H.

## Author contributions

N.R. and V.H. conceived the project and received critical input from S.S. and D.N. N.R. carried out most of the experiments and several  experiments were performed by E.D. and C.C. or individual experiments by A.M., J.E.S., S.B., J.K., A.J., R.F. and G.B., respectively. A.H., K.R., J.H., H.L., E.L., D.N. and S.S. contributed or supervised some of the analyses and R.F., U.D. and R.W. contributed reagents and advice. N.R. and V.H. wrote the manuscript.

## Additional information

**Competing interests:** The authors declare no competing financial interests.

