## [Peer Review File · Nature Communications]

Reviewers' comments:

Reviewer #1 (Remarks to the Author):

In this manuscript, Rehage et al. identify NUFIP2 as an interacting protein of Roquin1/2 and as a potential regulator of ICOS expression. The authors conducted an siRNA screen to identify regulators of ICOS expression, and followed up on one of the candidates, NUFIP2. The authors demonstrate that NUFIP2 engages in a protein-protein interaction with Roquin1/2, and that Roquin targets a double stem-loop motif in the 3'UTR of ICOS mRNA. However, the authors do not show how NUFIP2 influences ICOS expression (transcription, mRNA stability, protein translation?). Moreover, it is not clear if NUFIP2 acts on ICOS expression in a Roquin-dependent manner, and the mechanism by which NUFIP2 affects Roquin function (in case it acts in a Roquin-dependent manner) has not been solved.

Specific comments:

Fig.1: Authors should provide a full table with the outcome of the siRNA screen to document the results behind Fig.1e. In addition, one would like to know if more candidates were tested, and how many of them could be confirmed as (Roquin-dependent) regulators of ICOS expression.

Conceptually, I do not understand why this screen should only identify Roquin-dependent regulators of ICOS expression (as stated in the first sentence of the results section). Any siRNA-target that influences the ICOS reporter gene promoter, the synthesis or stability of ICOS protein, or even posttranscriptional mechanisms mediated by 3'UTR elements outside the Roquin-binding region, would also score positively in the screen. Please explain or choose appropriate wording.

Several of the data shown in the main part of the manuscript are not essential to the story, and can be moved to the supplement. E.g.:

Fig.2a,b (analysis of Staufien as a candidate that cannot be confirmed)

Fig.3d (scheme of the si-RNA resistant mutation)

Fig.3i: The authors claim that re-expression of NUFIP2 causes downregulation of ICOS expression only in cells expressing Roquin1/2 (- 4OH-TAM), but not in cells where Roquin1/2 is deleted (+ 4OH-TAM). I cannot see much difference between the - and + 4OH-TAM condition. If anything, there is a tiny bit of ICOS suppression by NUFIT2 re-expression in the absence of Roquin1/2 (+ 4OH-TAM, bottom panels), opposite of what the authors claim. Hence, I am not convinced that NUFIP2 acts in a Roquin1/2-dependent manner, nor is there evidence for any clear effect of NUFIT2 at all in this assay. This is a central point because the authors claim that NUFIT2 controls ICOS expression as a cofactor of Roquin. First, the authors should clarify at what level NUFIP2 affects ICOS expression. Does it influence ICOS mRNA stability (as one would expect if it acts via Roquin)? Proper mRNA decay experiments with half-life calculations need to be conducted. Second, the authors need to demonstrate convincingly that NUFIP2 acts on ICOS only in the presence of Roquin.

Fig.5 and 6: The binding assays demonstrate direct interaction between Roquin and NUFIT2, and reveal competition with a FMRP-NUFIT2 complex. For control, the authors should document how pure the bacterially expressed proteins are by showing coomassie-stained gels.

Fig. 7: Using EMSAs, the authors very nicely map the regulatory region, a double stem-loop motif, within the 3'UTR of ICOS mRNA that mediates high-affinity Roquin binding. In Fig.8, the authors further show that the middle domain of NUFIP2 has only a very weak RNA binding activity, yet that it can form a ternary complex with Roquin and the ICOS RNA. The authors should take this approach one step further by calculating precise Kd values for Roquin towards the ICOS RNA motif in the absence and presence of the NUFIT2 middle domain. If the authors measure higher affinity of Roquin in the presence of NUFIT2, this experiment could provide a mechanism by which NUFIT2

co-regulates ICOS expression. In case the middle domain of NUFIT2 does not alter the binding affinity of Roquin towards the ICOS RNA motif, the authors should consider testing whether full length NUFIT2 may have such an effect.

Fig. 8c: The model suggests that NUFIP2 recruits the decapping machinery. However, I cannot see any experimental evidence for this in the manuscript.

Reviewer #2 (Remarks to the Author):

The manuscript by Rehage et al, describes the identification of the NUFIP2 protein as a binding partner of Roquin. The authors show that Roquin and NUFIP2 are both required for regulation of ICOS mRNA. Finally, the authors identify a region in the ICOS 3' UTR that is required for regulation and binds Roquin and NUFIP2.

The manuscript contains several interesting observations but most of them have not been thoroughly investigated, in particular in the second half of the manuscript the authors have identified sequences in the ICOS 3' UTR that bind Roquin and NUFIP2 in vitro but mutations in these sequences have very minor effects on regulation and the significance of the binding observed in vitro remains unclear. Also the colocalization studies are not conclusive.

In my opinion the authors will be better off if they focus on the Roquin-NUFIP2 interaction and demonstrate the relevance of this interaction for ICOS mRNA regulation. An additional question that has not been addressed is whether NUFIP2 is required for regulation of other Roquin targets and if not, how specificity is achieved. The authors should minimally discuss why NUFIP2 binds Roquin only when associated with the ICOS mRNA.

Overall, the paper is interesting but not clearly written, it is difficult to understand the rationale of some of the experiments and how the experiments were done.

Additional comments:

1. The quality of the recombinant proteins used in the SRP measurements and gel shift assays should be demonstrated by SDS page.
2. It is not clear why NUFIP2 fragment comprising residues 255-411 was used in the gel shift assay.
3. Do the fragments of Roquin and NUFIP2 used in the gel shift assays interact with each other in vitro?
4. Figure 7i is not convincing. There is still regulation when mutant SM1, SM2 is expressed. How is this data normalized? A western blot showing equal expression of the proteins in all conditions is required.
5. In the IPs the percentage of the IP analyzed should be indicated. All Western blots lack molecular weight markers and the membranes are cut too close to the bands. Furthermore, it is not clear whether the IPs and inputs were analyzed in the same gel and processed in a similar manner (i.e., are the panels comparable?). If the samples were analyzed on the same gels why the figures show sliced bands.
6. Detailed information regarding the CRISPR knockout cell lines should be provided. Why is the NUFIP2 protein still detected?
7. In Figure 4 the authors show Roquin fragments and they claim they are N-terminal fragments but how was this determined?
8. In line 338 the authors claim that NUFIP2 increases the affinity of Roquin for the mRNA, but this conclusion is not warranted.
9. In Fig. 5a the input of the RNase+ sample is not shown.
10. In Fig. 7c and Fig. 7g, there is much less (or no) RNA in the last lane compared to the other lanes. In Figure 7g, the RNA seems to be degraded in the last lane.

11. The biacore curves in Fig. 6a are not as good as in Fig. 5g, and also the response units are lower (i.e., the sample with 4 μ M NUFIP2 should be equivalent in both experiments).

Reviewer #3 (Remarks to the Author):

In this manuscript Rehage and colleagues set out to identify novel cofactors of Roquin-1 and Roquin-2 using an RNAi screen of about 1500 genes functioning in posttranscriptional regulation. In the course of their study they identify NUFIP2 as a novel cofactor of Roquin-1/-2. NUFIP2 physically interacts with Roquin-1/-2. The NUFIP2 / Roquin-1 interaction allows binding to a tandem-stem loop structure in the far 3' end of the 3'UTR of ICOS mRNA.

The cell surface receptor ICOS plays an important role in the activation of T cells. The principles of post-transcriptional regulation of ICOS mediated by Roquin proteins are therefore of considerable interest to a broader audience. In addition, the study provides evidence that the combination of different RNA cis-elements contribute to the complexity and precision of posttranscriptional regulation. Of great interest is the observation that one RNA-binding protein can influence the binding affinity/specificity of another RNA-binding protein.

The manuscript is well written and clearly structured. Overall, this is a well-executed study with very convincing in vitro and in vivo data. Before publication, it would be helpful to clarify some issues:

1. The authors argue that their data indicate a cooperative function of the CDE-like tri-loop element and the unconventional tandem cis-element located hundreds of nucleotides downstream. In this context it would be helpful to conduct reporter studies with the isolated RNA recognition elements to show that the single elements show a reduced regulation whereas the combination of both is greater than the summed up regulation by each of the elements. Is the distance between the two 3'UTR elements of any relevance and if so what role does the position play. Do they have to be at different ends of the 3'UTR?
2. The authors suggest that Nufip2 increases the binding affinity of Roquin in a ternary complex with the ICOS cis-element. According to Figure 8b the binding difference is minor or not changed at all in the presence of Nufip2. The authors should quantify the affinity of Roquin alone and the presence of Nufip2 to the respective RNA substrate. Why is a 10fold molecular excess of Nufip2 necessary to accomplish the supershift. However, the interaction studies indicate that only a short segment of Nufip2 is interacting with Roquin N-terminus. The authors should discuss this observation, in particular when presenting a model (in Figure 8c) in which Nufip2 seems to make various different contacts to Roquin.

Minor issues:

- Quality of image presented in Figure 4c is very poor. It is nearly impossible to draw any conclusions
- Regarding Figure 4e. According to Fagerberg et al. (PMID 24309898) the highest expression of NUFIP2 is found in bone marrow and placenta. Similar expression levels are observed for ROQUIN.

Point-by-point response

We would like to thank all three reviewers for their appreciation of our manuscript as a “well-executed study” with “interesting observations” and “very convincing in vitro and in vivo data” that are “of considerable interest to a broader audience”. We are also grateful for their insightful comments that have helped us to improve on the mechanistic aspects of the story. We can now provide evidence that NUFIP2 acts on the stability of the ICOS transcript by determining an extended mRNA half-life in the absence of NUFIP2 and showing that this effect depends on the presence of ROQUIN (**Fig. 3h**). We extended our study of the mechanism of coregulation by demonstrating that the presence of NUFIP2 leads to a 3-fold better binding of Roquin/NUFIP-complexes to the RNA *cis*-element from the *ICOS* 3'-UTR (**Fig. 8c** and **Supplementary Fig. 7b**). Furthermore, we now show that Roquin not only cooperates with NUFIP2 in the recognition of *ICOS* mRNA, but both factors similarly recognize and cooperate in the binding of a *cis*-element that is responsible for the post-transcriptional regulation of the *Ox40* mRNA (**Fig. 8f** and **Supplementary Fig. 7c**). These new experiments therefore describe the molecular mechanism of cooperative binding and suggest that the cooperativity not only exists for the recognition *ICOS*, but also for *cis*-elements in other mRNA targets.

We have addressed all issues raised by the reviewers as follows:

Reviewer #1, specific comments

Fig.1: Authors should provide a full table with the outcome of the siRNA screen to document the results behind Fig.1e. In addition, one would like to know if more candidates were tested, and how many of them could be confirmed as (Roquin-dependent) regulators of ICOS expression. Conceptually, I do not understand why this screen should only identify Roquin-dependent regulators of ICOS expression (as stated in the first sentence of the results section). Any siRNA-target that influences the ICOS reporter gene promoter, the synthesis or stability of ICOS protein, or even posttranscriptional mechanisms mediated by 3'UTR elements outside the Roquin-binding region, would also score positively in the screen. Please explain or choose appropriate wording.

We provide the requested information in **Supplementary Table 1** and in the corresponding main text. We agree with this reviewer also in the second comment: The design of our screen did not exclude Roquin-independent regulators of ICOS expression. NUFIP2 was the only remaining candidate after our initial validation of 12 candidates that we performed by siRNA deconvolution (**Fig. 2** and **Supplementary Table 1**). The NUFIP2-mediated regulation proved to be Roquin-dependent later during hit validation and in the ability of NUFIP2 to bind to the *ICOS cis*-element. We have changed the wording in the main text accordingly.

Several of the data shown in the main part of the manuscript are not essential to the story, and can be moved to the supplement. E.g.:

Fig.2a,b (analysis of Staufen as a candidate that cannot be confirmed)

Fig.3d (scheme of the si-RNA resistant mutation)

We agree with this reviewer and have moved **Fig. 2a-c** and **3d** into **Supplementary Fig. 2 a-c** and **d**, respectively.

Fig.3i: The authors claim that re-expression of NUFIP2 causes downregulation of ICOS expression only in cells expressing Roquin1/2 (- 4OH-TAM), but not in cells where Roquin1/2 is deleted (+ 4OH-TAM). I cannot see much difference between the - and + 4OH-TAM condition. If anything, there is a tiny bit of ICOS suppression by NUFIP2 re-expression in the absence of Roquin1/2 (+ 4OH-TAM, bottom panels), opposite of what the authors claim. Hence, I am not convinced that NUFIP2 acts in a Roquin1/2-dependent manner, nor is there evidence for any clear effect of NUFIP2 at all in this assay. This is a central point because the authors claim that NUFIP2 controls ICOS expression as a cofactor of Roquin. First, the authors should clarify at what level NUFIP2 affects ICOS expression. Does it influence ICOS mRNA stability (as one would expect if it acts via Roquin)? Proper mRNA decay experiments with half-life calculations need to be conducted. Second, the authors need to demonstrate convincingly that NUFIP2 acts on ICOS only in the presence of Roquin.

Following the suggestions of this reviewer, we have performed mRNA decay experiments for *ICOS* in the presence and absence of NUFIP2 or ROQUIN-1 and -2 or in the absence of all three proteins (**Fig. 3g-h**). The *ICOS* mRNA half-life was prolonged 1.5 fold upon knockdown of NUFIP2 demonstrating that NUFIP2 regulates mRNA stability. Importantly, knockdown of Roquin-1/2 expression increased the half-

life to 2.2 fold, and a similar effect was obtained by combined NUFIP2/ROQUIN-1/2 knockdowns (2.3 fold). This indicates that the effect of NUFIP2 depends on the presence of ROQUIN.

Fig.5 and 6: The binding assays demonstrate direct interaction between Roquin and NUFIT2, and reveal competition with a FMRP-NUFIT2 complex. For control, the authors should document how pure the bacterially expressed proteins are by showing coomassie-stained gels.

We are now providing coomassie-stained gels for all proteins used in this study (**Fig. 5f** and **Fig. 6a**). These pictures show that the proteins are $\geq 95\%$ pure. Measured OD 260/280 values below 0.65 further indicated that the proteins were essentially free of nucleic-acid contaminations. We added a corresponding sentence to the methods section.

Fig. 7: Using EMSAs, the authors very nicely map the regulatory region, a double stem-loop motif, within the 3'UTR of ICOS mRNA that mediates high-affinity Roquin binding. In Fig.8, the authors further show that the middle domain of NUFIP2 has only a very weak RNA binding activity, yet that it can form a ternary complex with Roquin and the ICOS RNA. The authors should take this approach one step further by calculating precise K_d values for Roquin towards the ICOS RNA motif in the absence and presence of the NUFIT2 middle domain. If the authors measure higher affinity of Roquin in the presence of NUFIT2, this experiment could provide a mechanism by which NUFIT2 co-regulates ICOS expression. In case the middle domain of NUFIT2 does not alter the binding affinity of Roquin towards the ICOS RNA motif, the authors should consider testing whether full length NUFIT2 may have such an effect.

We thank the reviewer for this insightful comment. Following his/her recommendation, we have calculated K_d values from triplicate EMSAs. K_d values for Roquin towards the ICOS motif show a 3-fold increased affinity in the presence of NUFIP2. The results confirm our proposed mechanism by which NUFIP2 co-regulates the expression of Roquin targets through enhancing the affinity of Roquin towards RNA *cis*-elements. Furthermore, a new EMSA is provided, in which a supershift upon addition of NUFIP2 is better visible (**Supplementary Fig. 7a**). We also added arrowheads to the heights of shifted and supershifted bands, allowing the reader to easier grasp this point. These new insights provide additional strong evidence for the formation of a ternary complex consisting of ICOS RNA, Roquin and NUFIP2.

Fig. 8c: The model suggests that NUFIP2 recruits the decapping machinery. However, I cannot see any experimental evidence for this in the manuscript.

The model was designed to incorporate earlier results as well as recent findings from the literature, such as the direct interaction between NUFIP2 and Rck, which may be important for mRNA decapping (**Bish et al. 2015**). However, we agree with this reviewer that a very detailed model may provoke misunderstandings, therefore we are now providing a simplified model representing exclusively the findings of this manuscript in **Supplementary Fig. 7d**.

Reviewer #2 (Remarks to the Author):

The manuscript contains several interesting observations but most of them have not been thoroughly investigated, in particular in the second half of the manuscript the authors have identified sequences in the ICOS 3' UTR that bind Roquin and NUFIP2 *in vitro* but mutations in these sequences have very minor effects on regulation and the significance of the binding observed *in vitro* remains unclear.

We would like to respectfully point out that the correct interpretation requires the discrimination between the magnitude of ICOS reporter regulation achieved by either endogenous or overexpressed Roquin. Regulation by overexpressed Roquin has a high dynamic range and is gradually reduced with serial shortening of the ICOS 3'-UTR (**Supplementary Fig. 6b**). However, complete inhibition of regulation is only achieved with a full deletion of the 3'-UTR and regulation is only reduced by 50% when the 3'-UTR is shortened beyond the mapped tandem *cis*-element (**Supplementary Fig. 6b**). These findings are readily explained by the existence of additional low affinity interaction sites, which become sufficient for regulation, when Roquin expression is experimentally increased. In contrast, post-transcriptional regulation of ICOS by endogenous Roquin is only about two-fold, but this regulation is completely abolished when the 3'-UTR is shortened beyond the mapped tandem *cis*-element (**Fig. 7a**). Agreeing with this reviewer that the cellular model of induced deletion is more adequately representing the actual situation in the cell, we have repeated the reporter experiment in this system. We now provide a more comprehensive analysis of individual and combined mutants from former **Fig. 7h, i** in the

Rc3h1^{fl/fl};Rc3h2^{fl/fl};Cre-ERT2 MEF cell line that is used in **Fig. 7a**. Now, the regulation is completely reduced, similar to deletion of the 3'-UTR, when we introduce point mutations that abolish the formation of all three structural elements including the CDE-like, the non-canonical tri-loop and the octa-loop structures. Similar results are obtained when we combine mutations of two of these elements (CDE-like and tri-loop, CDE-like and octa-loop or tri-loop and octa-loop) (**Fig. 7h, i**), while individual mutations were only effective for the tri-loop and less effective for CDE-like and octa-loop. These data indicate that all three structures contribute to the regulation of the ICOS 3'-UTR by endogenous Roquin, which however fully depends on the tandem element that we have studied in more detail in this manuscript.

Also the colocalization studies are not conclusive.

We agree that due to the small size and low resolution, the colocalization studies were not adequately represented. We are now showing full-size images of our super-resolution microscopy data in **Supplementary Fig. 4 a, b** and will provide all images in highest resolution for final submission.

In my opinion the authors will be better off if they focus on the Roquin-NUFIP2 interaction and demonstrate the relevance of this interaction for ICOS mRNA regulation. An additional question that has not been addressed is whether NUFIP2 is required for regulation of other Roquin targets and if not, how specificity is achieved. The authors should minimally discuss why NUFIP2 binds Roquin only when associated with the ICOS mRNA.

We thank the reviewer for this fruitful comment. We now provide evidence that NUFIP2 cooperates with Roquin in the regulation of *ICOS* mRNA decay (**Fig. 3h**). To address the question about additional targets we have now tested also *Ox40* mRNA, and show that NUFIP2 enhances the binding affinity of Roquin also towards this mRNA target about four-fold. Interestingly, the binding motif in *Ox40* also contains tandem stem-loops (**Janowski et al. 2016**). Although NUFIP2 was described as an RNA-binding protein, we were unable to see specific interaction with the *ICOS* or *Ox40 cis*-element in the mRNA. We therefore propose that NUFIP2 either induces conformational changes in the Roquin RNA-binding domain or adds contacts of RNA- interaction in a complex with Roquin to enhance binding to low-affinity targets (represented in the model in **Supplementary Fig. 7d**). Considering that NUFIP2 itself shows low-affinity RNA binding itself the latter appears more likely to us. Regarding the last comment, we like to point out that NUFIP2 and Roquin do interact also in absence of RNA (**Fig. 5f-h** in the original and revised manuscript) with a K_d of 182 ± 72 nM).

Overall, the paper is interesting but not clearly written, it is difficult to understand the rationale of some of the experiments and how the experiments were done.

We have now provided additional explanations on all of the requested aspects. In light of reviewer 3 stating "The manuscript is well written and clearly structured", we would require more specific comments where to improve the writing in the manuscript even more.

Additional comments:

1. The quality of the recombinant proteins used in the SRP measurements and gel shift assays should be demonstrated by SDS page.

We are now providing coomassie-stained gels for all proteins used in this study. Measured OD 260/280 values below 0.65 further indicated that the proteins were essentially free of nucleic-acid contaminations. We added a corresponding sentence to the methods section.

2. It is not clear why NUFIP2 fragment comprising residues 255-411 was used in the gel shift assay.

NUFIP2 is in large parts predicted to be disordered. Attempts to express and purify full-length protein as well as various subfragments did not result in satisfying results. However, for NUFIP2 fragment 255-411 we obtained well-behaved protein of high purity (see **Fig. 5f**). It might be that the significant effects we observed with NUFIP2 255-411 in EMSAs are greater in the full-length context. However, in our hands this was impossible to test.

3. Do the fragments of Roquin and NUFIP2 used in the gel shift assays interact with each other in vitro?

Please see **Fig. 5f-h** in the original as well as in the revised manuscript. Our surface plasmon resonance experiments clearly demonstrate a direct interaction between these two protein fragments with a K_d of 182 ± 72 nM.

4. Figure 7i is not convincing. There is still regulation when mutant SM1, SM2 is expressed. How is this data normalized? A western blot showing equal expression of the proteins in all conditions is required.

We have replaced this figure by a new experiment showing the regulation of these and additional combined stem-loop mutants in Rc3h1/2^{fl/fl}; Cre-ERT2 MEF cells. In this cell line that we have established and validated earlier (**Jeltsch et al. 2014** and **Schlundt et al. 2014**) Roquin expression is ablated upon tamoxifen-treatment due to Cre-mediated excision of critical exons in the Roquin-encoding alleles. The same cell line was used for all constructs, and Roquin expression before and after 4' OH-tamoxifen-treatment is shown as a representative example in **Supplementary Fig. 6c**. As explained in the figure legend of **Fig. 7i, j** fold regulation was determined by dividing (ICOS MFI +4' OH-TAM)/(ICOS MFI -4' OH-TAM) and normalized to the regulation of ICOS wildtype .

5. In the IPs the percentage of the IP analyzed should be indicated. All Western blots lack molecular weight markers and the membranes are cut too close to the bands. Furthermore, it is not clear whether the IPs and inputs were analyzed in the same gel and processed in a similar manner (i.e., are the panels comparable?). If the samples were analyzed on the same gels why the figures show sliced bands.

Molecular weight markers are shown on the side of each blot. In most cases, the whole immunoprecipitate was loaded onto one lane of an SDS PAGE, and the membrane was cut horizontally into two halves for the analysis of two interacting proteins. When this was impossible due to similar size of the analyzed proteins, the IP reaction was split into two and loaded onto separate gels, and whole membranes were probed with individual antibodies. In the latter case, we have now indicated that each lane shows 50% of the immunoprecipitation sample.

6. Detailed information regarding the CRISPR knockout cell lines should be provided. Why is the NUFIP2 protein still detected?

We did not generate single cell clones, but worked with cells that were transduced with lentiCRISPR vectors and then selected for vector-encoded resistance. These bulk populations therefore are a mixture of cells containing 3, 6, 9 etc nucleotide indels that may still express the NUFIP2 protein, but the majority of the cells will have a frame shift due to 1, 2, 4, 5, 7 or 8 etc. nucleotide indels and are therefore NUFIP2 KO cells. However, this experiment has now been replaced by mRNA half-life determinations in the presence and absence of NUFIP2 without or with Roquin-1/2 knockdown (as suggested by reviewer 1).

7. In Figure 4 the authors show Roquin fragments and they claim they are N-terminal fragments but how was this determined?

This blot was probed with the Roquin-1- and Roquin-2-specific monoclonal antibody 3F12, which was generated in-house by immunizing rats with an amino-terminal fragment of Roquin-1. The specificity of this antibody was evaluated in detail (see **Vogel et al. 2013**) and allows to equally detect amino-terminal fragments of Roquin-1 and Roquin-2 in cell extracts that appear and accumulate due to cleavage of Roquin by the paracaspase MALT1 (see **Jeltsch et al. 2014**).

8. In line 338 the authors claim that NUFIP2 increases the affinity of Roquin for the mRNA, but this is conclusion is not warranted.

We have now calculated the K_d values from triplicate EMSAs. K_d values for Roquin towards the *ICOS* motif are 3-fold lower in the presence of NUFIP2 (**Fig. 8a-c**, and **Supplementary Fig. 7b**), and the same effect is observed with *Ox40*, another prominent Roquin target (**Figure 8d-f** and **Supplementary Fig. 7c**). These results confirm our proposed mechanism by which NUFIP2 coregulates the expression of Roquin targets though enhancing the affinity of the Roquin/NUFIP2 complex towards RNA *cis*-elements.

9. In Fig. 5a the input of the RNase+ sample is not shown.

The input lysate is the same for RNase+ and RNase- samples (HEK293T cells transfected with GFP-Roquin-1 and NUFIP2). It was therefore loaded only once onto the SDS PAGE gel.

10. In Fig. 7c and Fig. 7g, there is much less (or no) RNA in the last lane compared to the other lanes. In Figure 7g, the RNA seems to be degraded in the last lane.

We agree that the last lane of **Fig. 7c** was of suboptimal quality and now provide a better image. However, the EMSA in **Fig. 7g** consistently showed a smear over the entire lane at the highest protein concentration, whenever we repeated this experiment. We are therefore confident that this is an unspecific RNA-binding event at very high protein concentration and not RNA degradation. We decided to leave this lane in the figure to show at which high concentration (unspecific) RNA binding begins to occur. In order to avoid confusion, we added an explanatory note to the legend of **Fig. 7**: At the highest protein concentration unspecific RNA binding was observed, as observed by a broad smear of retarded RNA.

11. The biacore curves in Fig. 6a are not as good as in Fig. 5g, and also the response units are lower (i.e., the sample with 4 μ M NUFIP2 should be equivalent in both experiments).

We agree that the quality of the Biacore experiments in **Fig. 5g** is better than for **Fig. 6b**. This is an observation we consistently made with these samples and is also reflected in our general experience that some proteins behave ideally in Biacore and others do not. Despite this difference, we are convinced that the response curves in **Fig. 6b** are of good quality for the shown steady-state measurements of affinities. On the other hand, kinetic measurements with these curves (i.e. On and Off rates) should not be and have not been performed with this protein.

Response units (RU) are arbitrary values that depend on the amount of surface-coupled ligand, the size and distance of the analyte that binds to it and the fraction of active analyte/ligand. As long as there is not too much analyte coupled to the surface (i.e. thousands of RUs, where mass transfer effects might occur) the absolute amounts of RU have no influence on the measured binding parameters. Therefore, the different RU values provide no information on the quality or binding strength of an interaction and thus can be ignored here. We like to add that differences in RUs are always observed between different coupling reactions (i.e. independent experiments).

Reviewer #3 (Remarks to the Author):

1. The authors argue that their data indicate a cooperative function of the CDE-like tri-loop element and the unconventional tandem cis-element located hundreds of nucleotides downstream. In this context it would be helpful to conduct reporter studies with the isolated RNA recognition elements to show that the single elements show a reduced regulation whereas the combination of both is greater than the summed up regulation by each of the elements. Is the distance between the two 3'UTR elements of any relevance and if so what role does the position play. Do they have to be at different end of the 3'UTR?

To address these questions we have cloned and analyzed a new set of reporter constructs in their regulation by endogenous Roquin levels (**Fig. 7h-j**). We now show regulation of the reporter from a minimal response element (delC), in which all stem-loops are brought in close proximity (**Fig. 7j**). This construct as well as partial 5' and 3' deletions of sequences between the CDE-like and tandem loops (delA and delB) showed an increased regulation by Roquin, therefore the encoded distance has a negative impact, which may enable an additional level of regulation due to dynamic secondary structure formation. We also addressed the contribution of individual loops by combining two or three mutations of stem-loops and switched to another reporter cell-line (as requested by reviewer 2) now analyzing MEF cells that enable inducible deletion of Roquin-encoding alleles. Our results indicate that the non-canonical tri-loop (M3) is critical for the regulation by endogenous Roquin, but also combinations of mutations of the CDE-like and octa-loop as well as any other combination of loop mutations completely abolish this regulation. These data therefore support the cooperative effect of the proximal and the distal 3'-UTR elements.

2. The authors suggest that Nufip2 increases the binding affinity of Roquin in a ternary complex with the ICOS cis-element. According to Figure 8b the binding difference is minor or not changed at all in the presence of Nufip2. The authors should quantify the affinity of Roquin alone and the presence of Nufip2 to the respective RNA substrate.

We have now calculated precise K_d values from triplicate EMSAs. K_d values for Roquin towards the ICOS motif are 3-fold better in the presence of NUFIP2 (**Fig. 8b, c** and **Supplementary Fig. 7b**). We also performed the same experiment for a fragment of the *Ox40* 3'-UTR (**Fig. 8e, f** and **Supplementary Fig. 7c**) and made the same observation, thereby cross-validating the findings on *ICOS* binding and extending the potential target set of cooperative regulation.

Why is a 10fold molecular excess of Nufip2 necessary to accomplish the supershift. However, the interaction studies indicate that only short segment of Nufip2 is interacting with Roquin N-terminus. The authors should discuss this observation, in particular when presenting a model (in Figure 8c) in which Nufip2 seems to makes various different contacts to Roquin.

We have titrated the amount of NUFIP2 that is necessary to accomplish the supershift (**Supplementary Fig. 7a**). The supershift becomes evident at 180 nM NUFIP2, which is only a 2-fold molecular excess over Roquin. However, to be on the safe side, we chose 540 nM for subsequent experiments.

It is well possible that full-length NUFIP2 makes additional contacts and thus further strengthens this interaction. To test this possibility, we have tried to express and purify full-length NUFIP2. Unfortunately, we were unable to obtain recombinant full-length NUFIP2 with sufficient purity and quality. We therefore refrained from using such inferior protein samples for interaction studies, but mention the possibility of additional contacts between Roquin and full-length NUFIP2 in the Discussion.

Minor issues:

- Quality of image presented in Figure 4c is very poor. It is nearly impossible to draw any conclusions

We are now showing full-sized images in **Supplementary Fig. 4** and will happily provide all images in high resolution for final submission.

- Regarding Figure 4e. According to Fagerberg et al. (PMID 24309898) the highest expression of NUFIP2 is found in bone marrow and placenta. Similar expression levels are observed for ROQUIN.

We thank this reviewer for pointing out this interesting observation. We have performed western blots to compare Roquin and Nufip2 protein expression in mouse primary and secondary lymphoid organs. However, we have not included the data, since we were unable to detect high levels of both proteins in the bone marrow compared to splenocytes and thymocytes, which could either indicate a disconnect in RNA and protein abundance or the divergence of expression between mouse and human cells.

REVIEWERS' COMMENTS:

Reviewer #3 (Remarks to the Author):

The authors addressed all issues in depth.